# Evolutionary rescue of resistant mutants is governed by a balance between radial expansion and selection in compact populations

**Serhii Aif** [1,2], **Nico Appold** [1,2], **Lucas Kampman** [3,4], **Oskar Hallatschek** [3,4,5] ✉ **& Jona Kayser** [1,2] ✉

Mutation-mediated treatment resistance is one of the primary challenges for modern antibiotic and anti-cancer therapy. Yet, many resistance mutations have a substantial fitness cost and are subject to purifying selection. How emerging resistant lineages may escape purifying selection via subsequent compensatory mutations is still unclear due to the difficulty of tracking such evolutionary rescue dynamics in space and time. Here, we introduce a system of fluorescence-coupled synthetic mutations to show that the probability of evolutionary rescue, and the resulting long-term persistence of drug resistant mutant lineages, is dramatically increased in dense microbial populations. By tracking the entire evolutionary trajectory of thousands of resistant lineages in expanding yeast colonies we uncover an underlying quasi-stable equilibrium between the opposing forces of radial expansion and natural selection, a phenomenon we term inflation-selection balance. Tailored computational models and agent-based simulations corroborate the fundamental nature of the observed effects and demonstrate the potential impact on drug resistance evolution in cancer. The described phenomena should be considered when predicting multi-step evolutionary dynamics in any mechanically compact cellular population, including pathogenic microbial biofilms and solid tumors. The insights gained will be especially valuable for the quantitative understanding of response to treatment, including emerging evolution-based therapy strategies.

Many drug resistance mutations are associated with a decrease in growth rate in the absence of treatment, a phenomenon often referred to as the fitness cost of resistance, and thus subject to purifying selection[1–4]. In principle, slower-growing resistant clones can be rescued by acquiring subsequent compensatory mutations, offsetting their resistance-associated fitness cost[2,5–8] (Fig. 1a). However, the short lifetime and small size of less-fit intermediate clones, comprised of slower-growing cells, makes crossing such a fitness valley inherently rare, requiring large populations[9,10], environmental shifts[11], or long times[12] to become repeatedly observable at all.

[1]Max Planck Institute for the Science of Light & Max-Planck-Zentrum für Physik und Medizin, 91058 Erlangen, Germany. [2]Department of Physics, Friedrich-Alexander-University Erlangen-Nürnberg, 91054 Erlangen, Germany. [3]Department of Physics, University of California, Berkeley, CA 94720, USA. [4]Department of Integrative Biology, University of California, Berkeley, CA 94720, USA. [5]Peter Debye Institute for Soft Matter Physics, Leipzig University, 04103 Leipzig, Germany. ✉e-mail: ohallats@berkeley.edu; jona.kayser@mpl.mpg.de

**Fig. 1 | Synthetic mutation assay to study evolutionary rescue dynamics in expanding microbial colonies. a** Schematic of evolutionary rescue of a resistant mutant (red) associated with a fitness cost relative to its wild-type ancestor (gray) via a compensatory mutation that renders it resistant but without a cost (blue). **b** Outline of experimental competition assay of resistant clones (red, 10%) interspersed in a colony of susceptible wild-type cells (gray, 90%). **c** Microscope images of the colony front ≈ 1 h after inoculation (day 0). Scale bar 10 μm. **d** Narrow but persisting resistant clone at the front just prior to application of hygromycin (day 5). Scale bar 100 μm. **e** Resistant growth dome after initiation of hygromycin treatment (day 6). Scale bar 100 μm. Images in **c**–**e** are representative for experiments repeated in *n* = 6 independent colonies. **f** Schematic of synthetic mutation

system (see genotypes in Supplementary Table 1). A red fluorescing (RFP) and cycloheximide responsive (inactive *cyh2r*) genotype (top) switches to a cyan fluorescing (CFP) and cycloheximide unresponsive (active *cyh2r*) genotype (bottom) via loxP recombination using a *β*-estradiol-tunable Cre recombinase (Cre-EBD) (see Supplementary Movie 1). *UBQ* denotes a ubiquitin moiety that triggers proteolytic cleavage and *hygMX* indicates a constitutively expressed hygromycin resistance cassette. **g** Illustrating microscopy image of a colony one day after treatment application. Most resistant clones have been outcompeted by the wild-type (Extinction). Only one clone acquired a compensatory mutation (Rescue) and persisted to expand upon hygromycin application (Treatment failure).

We and others have recently demonstrated that in dense populations collective cell dynamics inherently decrease the power of selection by several orders of magnitude[13,14]. The ensuing alterations in evolutionary dynamics and clone longevity are likely to also affect evolutionary rescue. However, tracking the ongoing evolutionary trajectories of small individual clones is extremely difficult. Time-resolved deep sequencing approaches can identify emerging de novo subclones but lack the ability to spatio-temporally track them, especially in dense populations[15–17]. Microscopy-based clonal tracking approaches of fluorescently pre-labeled clones, while featuring exquisite spatio-temporal resolution, fail to capture newly arising mutant clones[13,14,18–20]. Consequently, we still lack empirical insight into how the altered evolutionary dynamics in dense populations impact the evolutionary rescue of slower-growing lineages and, as a result, drug resistance evolution. A better understanding of the fundamental processes governing evolutionary rescue in such crowded settings will be crucial to understand why drug resistance is so prevalent in many pathological cellular populations.

In this work, we study the evolutionary rescue of drug-resistant clones in densely-packed cellular populations via a genetically tailored yeast-based model system. Combining our empirical results with accompanying in silico models of colony growth and agent-based tumor simulations, we find that the competing effects of mechanically-driven radial range expansion and a clone-width-dependent reduction of selection pressures conspire to create a previously unidentified inflation-selection balance. The resulting stabilization of less-fit resistant clones drastically enhances their chance to acquire compensatory mutations, persist for long times and, upon drug application, become the seed for resurgent growth.

## Results

### Tracking evolutionary rescue dynamics via synthetic compensatory mutations

We aim at investigating the evolutionary rescue of slower-growing resistant clones, competing within a background of faster-growing but susceptible wild-type cells. In this context, evolutionary rescue refers to the de novo acquisition of a compensatory mutation that elevates the growth rate of resistant cells to that of the wild-type (Fig. 1a). A model system of multi-type yeast colonies, comprised of genetically tailored resistant and susceptible *S. cerevisiae* strains, allows us to track evolutionary rescue dynamics and the resulting fate of resistant lineages with high spatio-temporal resolution. Fluorescently labeled resistant cells, carrying a constitutive resistance against the drug hygromycin B, are interspersed at a low fraction (10%) into a population of hygromycin-susceptible wild-type cells (Fig. 1b). We then grow radially expanding colonies from a small droplet of this mixed inoculum, placed on a 2D agar substrate (see Supplementary Fig. 1). Resistant lineages, expanding from individual cells inoculated at the front, form well-segregated sectors that can be readily visualized via time-resolved fluorescence microscopy (Fig. 1c, d).

To emulate the cost of resistance, we selectively adjust the growth rates of resistant cells via the translational inhibitor cycloheximide, to which wild-type cells are insensitive[21]. Resistance-associated growth rate reductions have been reported to vary substantially (0−75%), depending on the type of cell, the drug, and the mechanism of resistance, with low-cost variants being most frequently observed in clinical isolates[2,22]. The moderately low fitness cost of $s = 0.013 \pm 0.006$ (50 nM cycloheximide), chosen for our main experiments, reflects this distribution while allowing for optimal data acquisition within the time and throughput constraints of our assay.

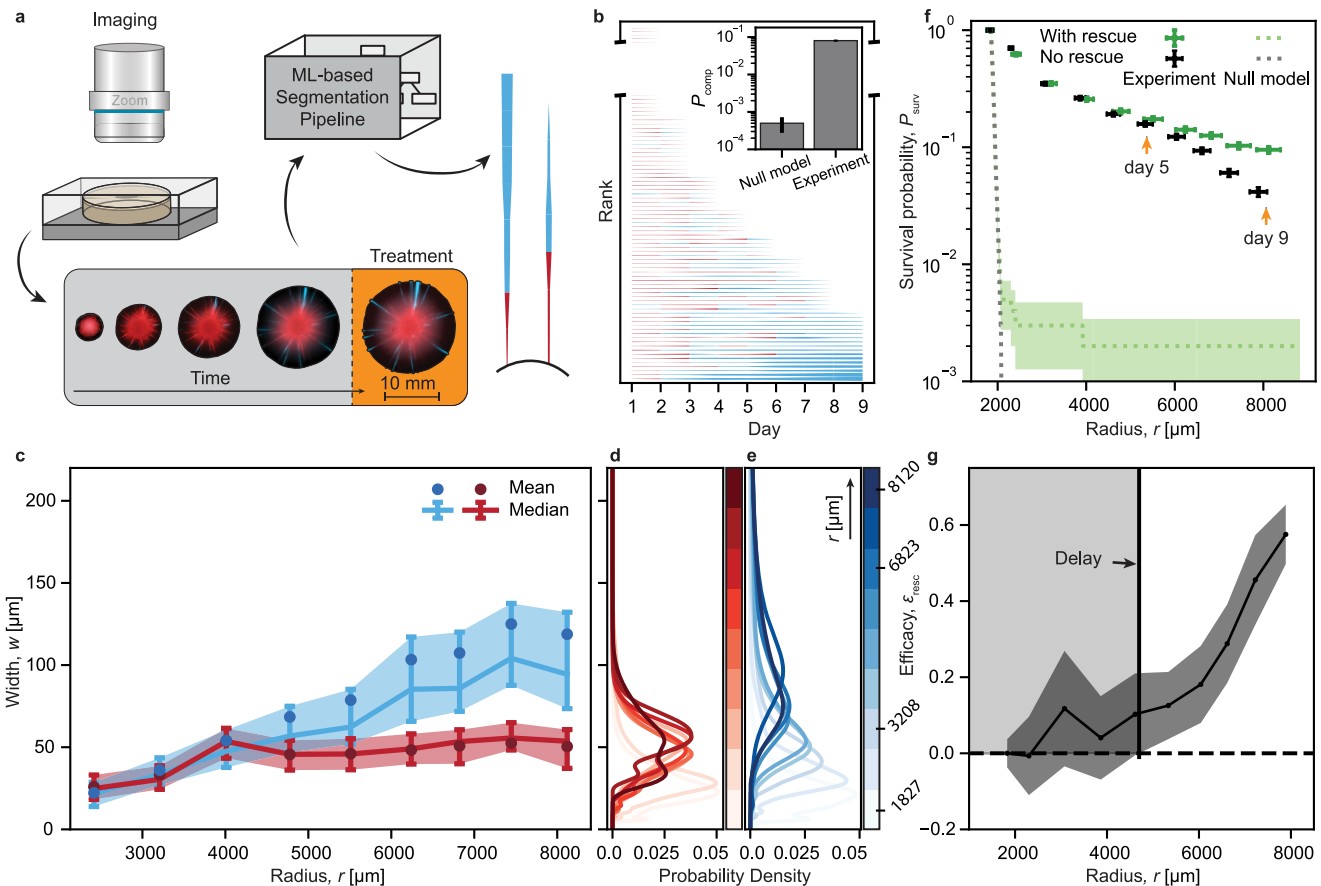

**Fig. 2 | Slower-growing resistant mutants are stabilized at a quasi-constant equilibrium width, reshaping the acquisition and effect of compensatory mutations. a** Experimental data analysis pipeline. Time series of colony images are acquired via fluorescence microscopy and analyzed via a machine-learning-based image segmentation and processing pipeline to extract clone trajectories. **b** Single-lineage trajectories (1 to 9 days of growth) for one representative colony (of $n = 48$ total number of colonies), sorted for trajectory length and type. Line width is proportional to the clone width at the colony edge. Not all clones extinct after one day are shown (see broken axis). Compare to Supplementary Fig. 3 for trajectories of no-rescue control. Inset: Probability $P_{comp}$ for null model simulations ($n_0 = 10,000$ clones) and experiments ($n_0 = 2898 \pm 126$ clones from $n = 18$ independent colonies) (see main text). **c** Width of compensated (blue) and uncompensated (red) clones. Connected points represent the median width with error bars/shaded areas indicating interquartile ranges. Circles represent mean values. $n = 18$ independent colonies with a mutation rate $\mu = 2.65 \pm 0.25 \times 10^{-4} \, \mu m^{-1}$ (4 nM $\beta$-estradiol). **d**, **e** Estimated probability densities for the width of uncompensated (red) or compensated (blue) clones, respectively (see "Methods" for details). Hue indicates increasing colony radius over time. **f** Clone survival probabilities at the front (Eq. (1)) for experiments with compensatory mutations (with-rescue, $\mu = 2.65 \pm 0.25 \times 10^{-4} \, \mu m^{-1}$, 4 nM $\beta$-estradiol) and minimal mutations (no-rescue, $\mu = 5.6 \pm 3.5 \times 10^{-6} \, \mu m^{-1}$, 0 nM $\beta$-estradiol) (points), and null model (doted lines). $n_0 = 2898 \pm 126$ clones in $n = 18$ independent colonies for both with-rescue and no-rescue experiments. See Supplementary Fig. 4 for survival probabilities with different mutation rates. Error bars/shaded area indicate Poisson distribution SD (vertical axis) and SD of the mean (horizontal axis). Arrows indicate treatment times as in Fig. 3. **g** Efficacy of compensatory mutations (Eq. (2)). Shaded areas indicate propagated SDs. Gray box represents the window of inefficacy, during which efficacy remains zero within errors. Source data are provided as a Source data file.

In the context of this study, cycloheximide is solely used to adjust the relative fitness between resistant and susceptible clones, while the resistance itself refers to hygromycin B. In addition, it should be noted that quantitative measures of the relative fitness cost refer to the rate of change in clone width, rather than the doubling rate of the comprising cells (see Supplementary Fig. 2). As a result of the fitness cost of resistant clones, we observe their width at the expanding front to remain small (~10 cells) during an initial hygromycin-free growth phase (Fig. 1d). Application of hygromycin halts the growth of susceptible wild-type cells while resistant clones still present at the colony edge continue to expand outward in a semi-circular pattern (Fig. 1e). We will refer to these rapid expansions of resistant lineages after drug application as resurgent growth domes. It should be noted that the specific form of treatment failure is likely to be different in other contexts, such as the antibiotic treatment of a biofilm or the effects of chemotherapy on a solid tumor. For example, dead yeast cells retain their structural integrity upon treatment and do not free up any space.

To additionally detect compensatory mutation events and track the evolutionary trajectories of emerging de novo compensated subclones, we introduce a recombinase-based synthetic mutations system (Fig. 1f). Slower-growing resistant cells can stochastically switch at a rate $\mu$ from a slower-growing, red-fluorescent initial state to a cyan-fluorescent rescued state which has a growth rate matching that of the non-switching wild-type cells. The mutation rate $\mu$ is set by the concentration of $\beta$-estradiol (Supplementary Fig. 2). Unless otherwise indicated, mutation rates were set to $\mu = 2.65 \pm 0.25 \times 10^{-4} \, \mu m^{-1}$ to obtain a sufficient number of evolutionary rescue events within the experimental time frame while maintaining a low probability of clonal interference of multiple compensated subclones within one resistant lineage. Note that both uncompensated and compensated cells carry the hygromycin resistance.

These synthetic mutations allow us to monitor incoming compensatory mutations throughout colony expansion and directly measure their effects on long-term persistence and treatment failure. With each initial clone representing an independent experiment, this high-throughput approach allowed us to quantitatively assess the fate of several thousand resistant lineages in parallel. Figure 1g shows a post-experiment image of a colony with clone boundaries as a fossil record

of past clone widths. We find that slower-growing resistant clones form narrow yet surprisingly persistent streaks before eventually being expelled from the expanding front. A compensated resistant subclone, in contrast, can persist long-term, even grow in size, and eventually seed a resurgent growth dome upon hygromycin treatment.

## The acquisition and effect of compensatory mutations is governed by a stabilization of uncompensated clones

To quantify the underlying evolutionary dynamics, we recorded the complete history of $N \sim 10{,}000$ individual resistant clones via time-resolved, multi-scale fluorescence microscopy in 24 h intervals (Fig. 2a). Clones were initially associated with an intrinsic fitness cost of $s = 0.013 \pm 0.006$ but could acquire a compensatory mutation at rate $\mu$ (see figure captions).

Segmenting images via a machine-learning-based, pixel-wise segmentation pipeline, we measured the width and compensation state (uncompensated (·) or compensated (+)) of clones at the colony edge to reconstruct their complete evolutionary trajectory (see Fig. 2b).

To quantify the prevalence of compensatory mutations, we calculated the probability $P_{\mathrm{comp}} = n^{+}/n_0$ of a clone to acquire a compensatory mutation and survive until the end of the experiment. We then compared our experimental results to a baseline obtained from a minimal in silico null model of radial range expansion. In this model, the trajectories of individual sector boundaries are simulated as selection-biased 1D random walks on a radially expanding periphery (see below for a detailed description). In brief, boundaries undergo a random step in angular space for each step of radial expansion. Competition between adjacent clones is implemented by shifting the Gaussian kernel from which the angular step size is drawn. This approach has been previously demonstrated to be well-suited to capture many key aspects of evolution and range expansion in spatially-structured populations, such as local competition, gene surfing, and the effects of radial growth[19,23,24]. Yet, our experiments show a 10-fold higher prevalence of compensated clones in comparison to this null model (Fig. 2b, inset).

Our assay also gives direct access to the width distributions of uncompensated and compensated clones (Fig. 2c–e). For compensated (blue) clones, we find that the mean width $w$ continuously grows with increasing radius while the underlying width distribution gradually widens (Fig. 2c, e). The mean width of uncompensated (red) clones, however, initially rises but then saturates for $r \gtrsim 4000\,\mu\mathrm{m}$ and remains constant at an equilibrium width $w_{\mathrm{eq}} = 50 \pm 17\,\mu\mathrm{m}$. This stabilization is also reflected in the width distribution of uncompensated clones, which remains narrowly confined around $w_{\mathrm{eq}}$ (Fig. 2d).

The observed width equilibrium should also have a marked impact on lineage fate, prolonging overall sector survival. We also expect that clones should become extinct at a constant rate that is set by the equilibrium width. In the absence of compensatory mutations, this should result in a gradual exponential decay of the survival probability $P_{\mathrm{surv}}$, while in the presence of compensatory mutation we would expect $P_{\mathrm{surv}}$ to level off at a non-zero value.

To test this hypothesis, we measured the radius-dependent survival probability

$$P_{\mathrm{surv}}(r) = \frac{n(r)}{n_0} \qquad (1)$$

with $n(r)$ and $n_0 = n(r = r_0)$ denoting the respective number of lineages present at the front when the colony has expanded to a radius $r$ (Fig. 2f).

Comparing survival probabilities of experiments to null model simulations for both a no-rescue ($\mu < 10^{-5}\,\mu\mathrm{m}^{-1}$) and a with-rescue ($\mu = 2.65 \pm 0.25 \times 10^{-4}\,\mu\mathrm{m}^{-1}$) scenario, we find that experimentally observed survival probabilities exceed those of the respective null

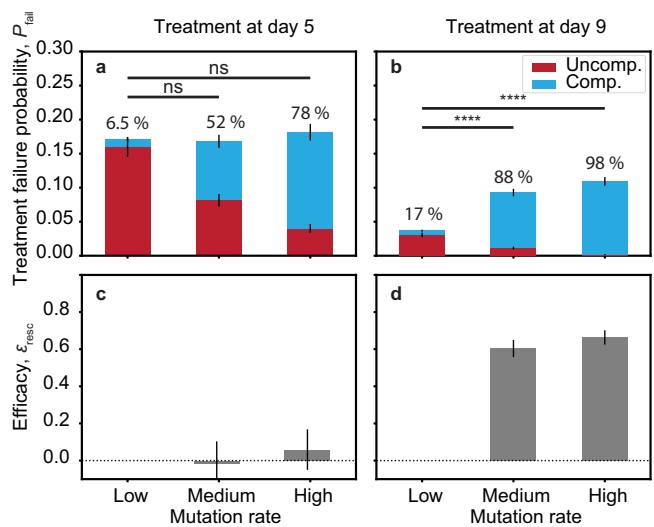

**Fig. 3 | The impact of compensatory mutations on treatment failure is delayed.** **a**, **b** Treatment failure probabilities of experiments with different mutation rates for early (**a**, day 5, $n = 6$ independent colonies for each mutation rate) and late (**b**, day 9, $n = 18$ independent colonies for each mutation rate) hygromycin treatment (Eq. (3)) as measured via the number of resurgent growth domes. The fitness cost of all experiments was $s = 0.013 \pm 0.006$ (Supplementary Fig. 2a, b). Mutation rates were either low ($\mu = 5.6 \pm 3.5 \times 10^{-6}\,\mu\mathrm{m}^{-1}$), medium ($\mu = 2.65 \pm 0.25 \times 10^{-4}\,\mu\mathrm{m}^{-1}$), or high ($\mu = 4.48 \pm 0.34 \times 10^{-4}\,\mu\mathrm{m}^{-1}$) (Supplementary Fig. 2c, d). Colors represent the fraction of compensated (red) and uncompensated (blue) growth domes. Numbers refer to the percentage of compensated clones. The difference between the low mutation rate control and experiments with compensatory mutations are non-significant for treatment after day 5 (**a**, $p$-values 0.58 and 0.22 for medium and high switching rates, respectively) but very significant for treatment after day 9 (**b**, $p$-values $4 \times 10^{-22}$ and $7.5 \times 10^{-30}$), one-sided t-tests. Errors of $P_{\mathrm{fail}}$ indicate one SD of Poisson distribution. **c**, **d** Efficacy of evolutionary rescue with reference to low mutation experiment for treatment at day 5 or day 9, respectively. Error bars indicate propagated SDs. Source data are provided as a Source data file.

model data by at least one order of magnitude at the end of our experiments. The difference is especially drastic for the no-rescue scenario in which the null model did not yield any surviving clones for $r > 2100\,\mu\mathrm{m}$ while experiments exhibit a long-tailed, exponential-like decay. This behavior is consistent with our observation of a steady equilibrium width.

Intriguingly, for $r \lesssim 5000\,\mu\mathrm{m}$ we find experimental survival probability of the no-rescue scenarios to be virtually indistinguishable from those measured in with-rescue samples. In the later phase of expansion ($r \gtrsim 5000\,\mu\mathrm{m}$), however, the with-rescue survival probabilities level off while those of the no-rescue counterpart continue to decay. This divergence can be quantified by calculating the efficacy of rescue as (Fig. 2g)

$$\mathcal{E}_{\mathrm{resc}} = 1 - \frac{P_{\mathrm{surv}}^{\mathrm{no\text{-}rescue}}}{P_{\mathrm{surv}}^{\mathrm{with\text{-}rescue}}} \qquad (2)$$

Here, $\mathcal{E}_{\mathrm{resc}} = 0$ indicates no observed difference between both scenarios while $\mathcal{E}_{\mathrm{resc}} = 1$ represents the limit of all surviving clones to be exclusively in the with-rescue sample. In our experiments, efficacy remains zero within errors for $r < 4700\,\mu\mathrm{m}$, after which it continuously rises to $\mathcal{E}_{\mathrm{resc}} = 0.6 \pm 0.1$ at the end of experiments.

## The impact of evolutionary rescue is delayed

The observed delay in efficacy generates an initial window of inefficacy during which evolutionary trajectories seem unaffected by evolutionary rescue. Notably, the extent of this delay seems temporally decoupled from the acquisition of compensatory mutations, the majority of which having already occurred before $r = 3000\,\mu\mathrm{m}$

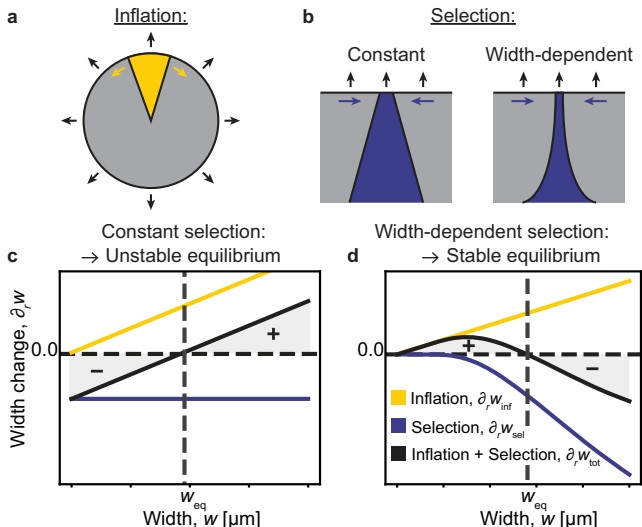

**Fig. 4 | The opposing effects of peripheral inflation and width-dependent selection create a quasi-stable equilibrium width. a** Illustration of inflation of sector width due to radial population expansion. **b** Illustration of constant and width-dependent negative selection in an inflation-free flat front. **c**, **d** Schematic graphs of lateral width change per radial colony expansion, $\partial_r w$, for constant (**c**) and width-dependent (**d**) selection scenarios, respectively. Yellow lines indicate the contribution of inflation for a given radius $r$, blue lines indicate the contribution of effective (constant or width-dependent) selection, and black lines represent the sum of both. Note that the slope of the yellow inflation line will gradually decrease as the colony expands. Shaded areas indicate inflation dominated (+, growing clones) or selection dominated (−, shrinking clones) width regions. See Supplementary Figs. 7 and 8 for width dynamics.

(Supplementary Fig. 5). Together, this suggests an interesting consequence: The probability of treatment failure should be independent of evolutionary rescue if therapy is initiated within the window of inefficacy.

To test this prediction, we conducted a series of therapy mimicry experiments, initiating treatment either within or substantially after the window of inefficacy. In these experiments, we expanded colonies for an initial pre-treatment phase at different mutation rates for either 5 or 9 days (see arrows in Fig. 2f). We then initiated treatment with hygromycin B, halting wild-type growth, and counted resurgent growth domes after one day of post-treatment regrowth. Analogous to $P_{surv}$, the respective treatment failure probabilities $P_{fail}$ for each scenario is then given by

$$P_{fail} = \frac{N_{fail}}{n_0} \qquad (3)$$

where $N_{fail}$ is the number of growth domes and $n_0$ is again the number of initially inoculated clones. We further measured the proportion of the total treatment failure probability attributed to either compensated (blue) or uncompensated (red) clones (Supplementary Fig. 6).

Comparing the treatment failure probabilities of different mutation rates, we find that the fraction of compensated growth domes increases with mutation rate (Fig. 3a). Intriguingly, this shift in compensation status does not directly translate to a change in overall treatment failure probability, which remains essentially unaffected for the early treatment point. Delaying treatment until day 9, in contrast, yields a significant difference in treatment failure probabilities between control and samples with increased mutation rate (Fig. 3b). The contrast between early and late treatment time points can also be appreciated by comparing the respective efficacy values $\mathcal{E}_{resc} = 1 - P_{fail}^{no-rescue}/P_{fail}^{with-rescue}$ (Fig. 3c, d).

## The occurrence and impact of evolutionary rescue is governed by a stable inflation-selection balance

Motivated by the observed width stability of uncompensated clones, we tested our hypothesis that this phenomenon might also be a key determinant of evolutionary rescue dynamics, including the window of inefficacy. The origin of the plateau in width can be rationalized by considering the interplay of the two main forces driving the change of mean clonal width, (i) global inflation of the population front due to radial growth (Fig. 4a) and (ii) natural selection (Fig. 4b). For a slower-growing clone, these forces are opposing each other with inflation increasing clone size and selection decreasing it, so that the total width change per radial expansion is

$$\partial_r w_{tot} = \partial_r w_{inf} - \partial_r w_{sel} \qquad (4)$$

with the clonal width $w$ measured as the arc length between delimiting sector boundaries. In the simplest scenario, inflation depends on the sector width and the current population radius as $\partial_r w_{inf}(w,r) = w/r$ while selection only depends on the relative fitness difference $s$ of the adjacent clones $\partial_r w_{sel} = \sqrt{|s|(2+s)}$. Such a constant selection scenario has been comprehensively investigated in previous studies of range expansions[18–20,25]. Even though a constant selection scenario can yield an equilibrium clone width $w_{eq}$ at which inflation and selection forces cancel each other out ($\partial_r w_{tot}(w_{eq}) = 0$), this equilibrium is inherently unstable (Fig. 4c). Clones of width $w < w_{eq}$ are selection dominated and continue to shrink while those of larger width $w > w_{eq}$ are inflation dominated and expand. However, we and others have recently demonstrated that in dense populations, similar to those investigated here, selection decreases with clone width[13,14]. For such a width-dependent selection $\partial_r w_{sel}(w)$, a stable equilibrium width can exist, such that small clones are inflation dominated while bigger clones are selection dominated (Fig. 4d). Note that the strength of inflation is inversely proportional to the colony radius $r$. As a result, $w_{eq}$ will gradually increase for a constant selection scenario and decrease for width-dependent selection.

To quantitatively test this inflation-selection balance hypothesis, we simulated both constant and width-dependent selection scenarios, modeling the trajectories of individual sector boundaries as biased 1D random walks on a radially inflating surface (Fig. 5a)[13,24]. We found that our experimentally observed survival probabilities could not be captured by any constant fitness scenario, even if effective selection was reduced by several orders of magnitude (Fig. 5f, dashed lines; see also Supplementary Figs. 9 and 10).

However, using a width-dependent selection coefficient $s_{eff}(w)$ (Fig. 5b, inset) not only resulted in a plateauing and narrowly distributed width of uncompensated clones (Fig. 5c–e) but also accurately reproduced empirical survival probabilities and efficacies (Fig. 5f, g, solid lines). Note that the above rationale for an inflation-selection balance is robust across a wide spectrum of width dependence, solely requiring that $\partial_r w_{sel}(w)$ approaches zero more rapidly than inflation for declining width (see Supplementary Fig. 11). Here, we used a logistic form of $s_{eff}(w) = s \cdot 2/(1 + e^{w_c/w})$ as a minimal heuristic model with only one free parameter (the critical width $w_c$).

We previously demonstrated that such a clone-width-dependent reduction in effective selection can inherently emerge as a result of collective cell dynamics in dense cellular populations (see Fig. 2g in ref. [13]). In short, distance-dependent mechanical coupling of cell motion prevents the differential displacement required for selection to act. The exact form of this width-dependence, and with it the values of $w_{eq}$, may differ substantially between systems. However, the fundamental concept of inflation-selection balance and its evolutionary consequences may extend to other dense populations, including pathogenic bacterial biofilms or solid tumors, that exhibit the minimal set of necessary ingredients: (i) the inflation of a peripheral growth layer and (ii) width-dependent selection.

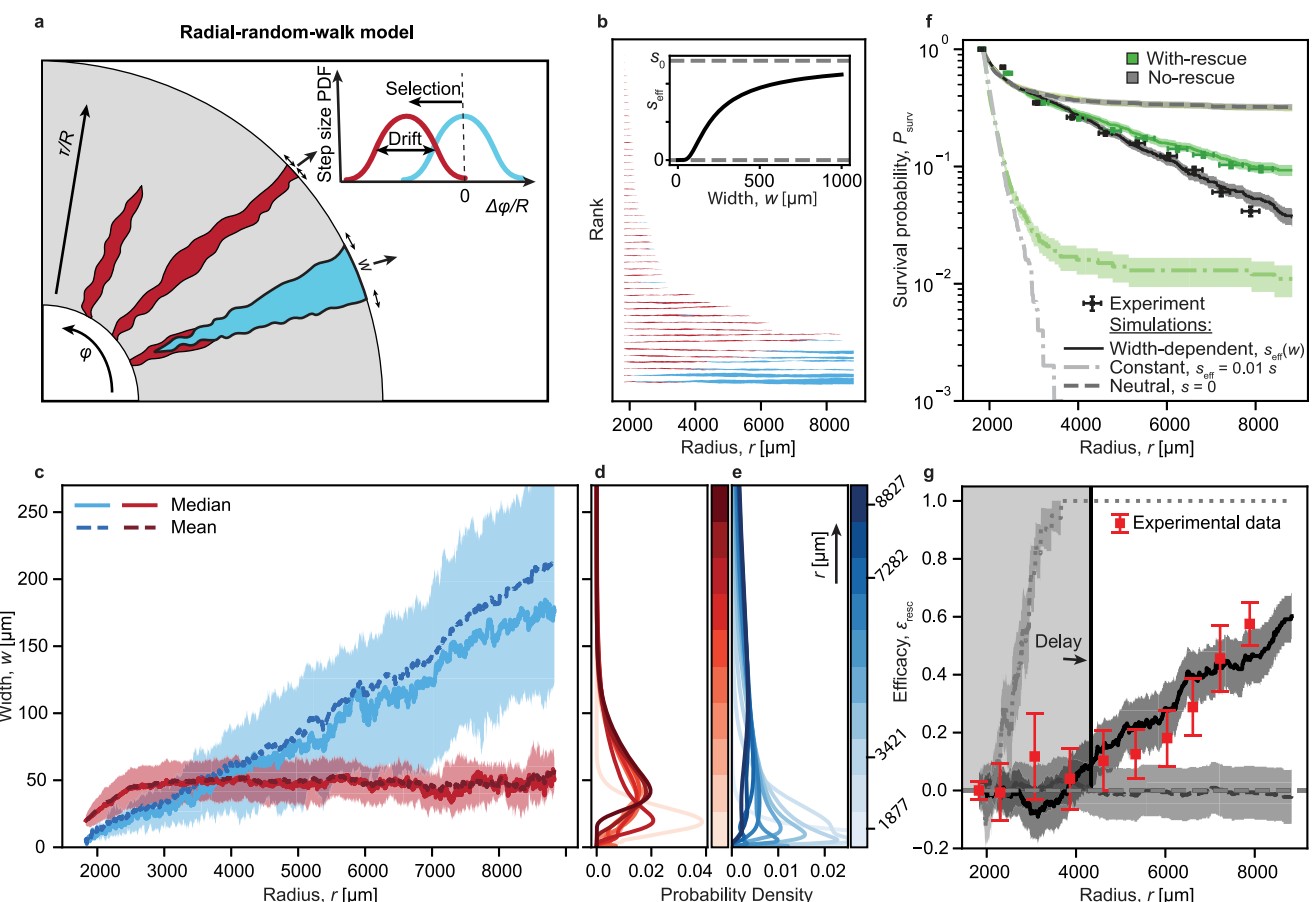

**Fig. 5 | Including inflation-selection balance in a random walk model of range expansion reproduces experimental observations. a** Schematic of radial-random-walk model, simulating clone boundaries as a selection-biased 1D random walk along a radially expanding periphery. Inset: Selection can either reflect the full fitness cost (null model, see Fig. 2b, f), be reduced by a constant factor (**f**) or change as a function of clone width (**b**–**g**). **b** Simulated trajectories (same radius range as in experiments). Line width is proportional to the clone width at the colony edge. Inset: Width-dependent effective selection coefficient (see main text). **c** Width of compensated (blue) and uncompensated (red) clones. Solid lines represent the median width and shaded areas indicate interquartile ranges. Dashed lines represent mean values. **d**, **e** Estimated probability densities for the width of uncompensated (red) or compensated (blue) clones (see "Methods" for details). Hue indicates increasing colony radius over time. **f** Clone survival probabilities at the front (Eq. (1)) for different simulation scenarios (see legend). Note that with-rescue and no-rescue data overlaps for the neutral scenario. Shaded area indicates Poisson distribution SD. Experimental data is identical to that shown in Fig. 2f. **g** Efficacy of compensatory mutations (Eq. (2)) for the same scenarios shown in panel **f**) with matching line styles. Shaded areas indicate propagated SDs. Gray box represents the window of inefficacy for the width-dependent selection scenario. Experimental data (red) is identical to that shown in Fig. 2g. Source data are provided as a Source data file, if applicable (see "Data availability" statement).

## Evolutionary rescue and treatment failure in an in silico tumor model

To assess the relevance of our findings in the context of solid tumors, we conducted agent-based simulation of tumor growth using a tailored implementation of the PhysiCell platform[26]. In short, cells grow and divide in an explicitly simulated nutrient microenvironment and repulsively interact via a distance-dependent force, resulting in non-motile, overdamped motion (Fig. 6a). In our implementation, individual cells can additionally mutate at a predetermined stochastic rate and fitness effect. Cells in the population interior stop proliferating due to a lack of nutrients, creating a peripheral growth layer.

Similar to our experimental assay, we simulated the expansion of 2D in silico tumors starting from a mixture of resistant but slower-growing cells interspersed at a low fraction into a background of faster-growing wild-type cells.

Resulting in silico tumor populations inherently capture the full finger print of inflation-selection balance. Following an analysis pipeline equivalent to that applied to experimental data (Fig. 6b), we again find stabilized trajectories of uncompensated clones, the characteristic long tail of no-rescue survival probabilities, and the ensuing delay in efficacy of evolutionary rescue (Fig. 6c–g).

Leveraging the spatio-temporal resolution of our simulations and the ability to precisely control timing and position of compensatory mutations, we find that the life history of a clone can be divided into four characteristic phases: (i) establishment phase, (ii) equilibrium phase, (iii) transition phase, and (iv) escaped phase (Fig. 6h). At the beginning of the initial establishment phase, uncompensated clones are very small by definition, similar to a de novo resistant clone just after acquiring the resistance-conveying mutation. During this phase, the probability of a clone to be driven to extinction by random genetic drift is high.

Those clones that do not succumb to genetic drift right away enter the equilibrium phase, in which the width of a clone remains constant due to the opposing forces of natural selection and peripheral inflation. In our simulations, we find this finite equilibrium width to be $39.24 \pm 22.32\,\mu m$, or $w_{eq} = 3.27 \pm 1.86$ cell diameters. While clones in this phase can still fluctuate to extinction, the rate is drastically reduced in comparison to the initial establishment phase due to an increased mean clone width. This stabilization at a small but finite clone width can now be interpreted as the root cause for the narrow yet persistent streaks of uncompensated clones observed in Fig. 1g.

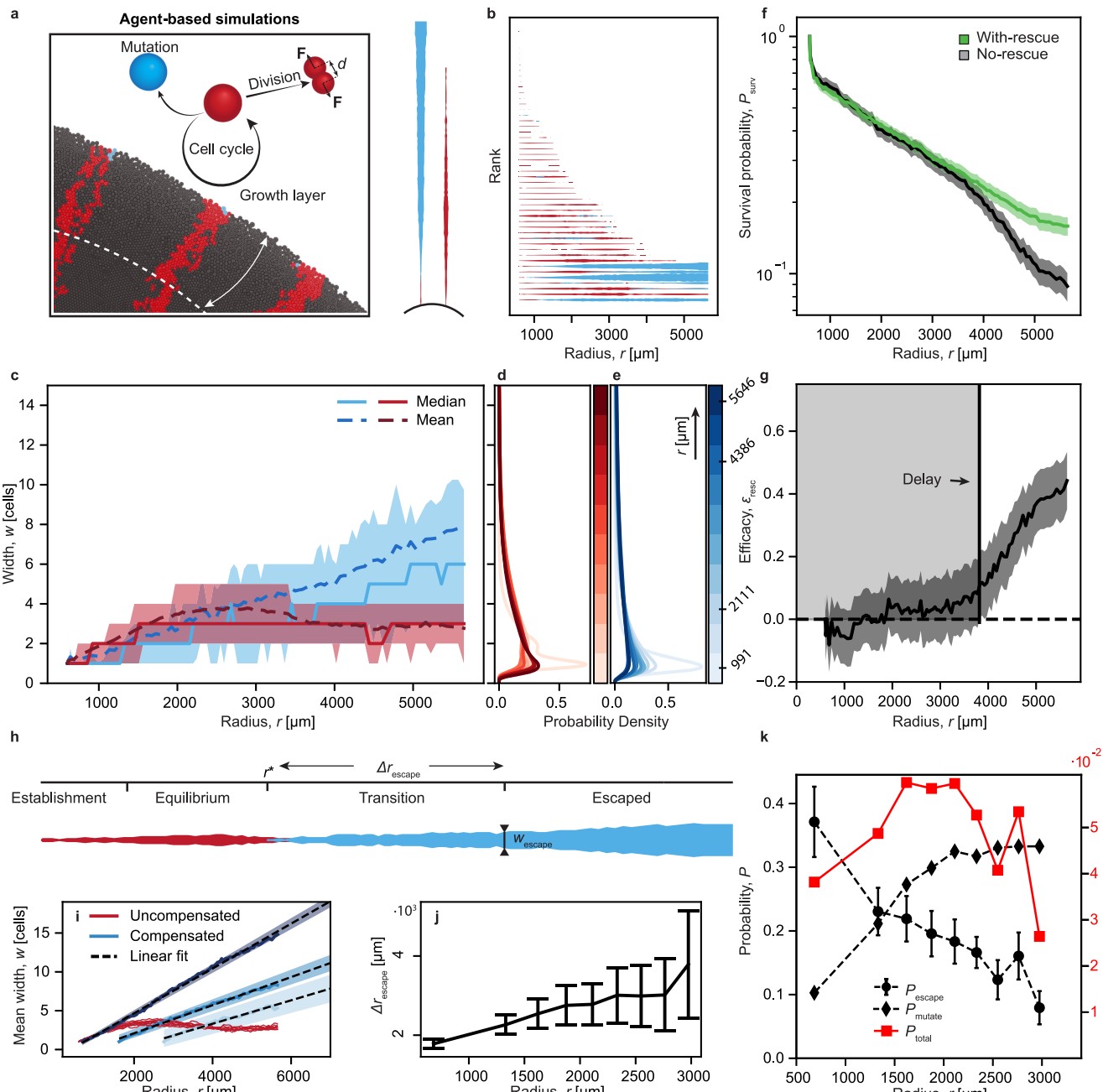

**Fig. 6 | The interplay of inflation-selection balance and evolutionary rescue inherently emerges in an agent-based in silico tumor model. a** Schematics of the simulation set-up (see "Methods" section for details). **b** Simulated trajectories. Line width is proportional to the clone width at the colony edge. **c** Width of compensated (blue) and uncompensated (red) clones. Solid and dashed lines represent the median and mean widths, respectively, and shaded areas indicate interquartile ranges. $s = 0.21$ (see Supplementary Fig. 12 for different fitness costs). **d, e** Estimated probability densities (see Fig. 5d, e and "Methods" for details). **f** Clone survival probabilities at the front (Eq. (1)). Shaded area indicates Poisson distribution SD. **g** Efficacy of compensatory mutations. Shaded areas indicate propagated SDs. Gray box represents the window of inefficacy. **h** Representative simulated clone trajectory exhibiting all phases (see main text). **i** Mean clone width development of

uncompensated clones (red lines) and mutations triggered at different colony radii $r^*$ (blue solid lines). Lines begin at their mutation time point. Hue decreases with the mutation start. Dashed lines are linear fits to the width of compensated clones. Shaded areas represent SD of the fitted lines. **j** Radial expansion $\Delta r_{escape}$ from the point of mutation $r^*$ to escape $w_{escape}$ as function of $r^*$ (mean ± SD). **k** Probability $P_{escape}$ (black circles) to grow above escape width $w_{escape} = 6$ cells in 2500 µm of radial colony growth after mutation. Error bars indicate one SD assuming a Poisson distribution. Probability $P_{mutate}$ (black diamonds) to get a mutation (with mutation rate $\mu = 0.1$ per step) at the given radius. Combined probability $P_{total}$ (red squares) for a clone to acquire a mutation and eventually grow to the width $w > w_{escape}$. $n = 20$ simulated colonies for each $r^*$. Source data are provided as a Source data file.

The acquisition of a compensatory mutation then initiates a transition phase during which the survival probability gradually transitions from low to high. Even though compensated mutant cells now double at the rate of wild-type cells and even have a fitness advantage over their neighboring uncompensated ancestor, de novo

compensated clones originate from a single cell and are therefore very small initially. At these small widths, the benefits of positive selection and radial inflation are small in comparison to the effects of random width fluctuations. As a result, a substantial fraction of nascent compensated subclones will be driven to extinction, similar to the fate of

uncompensated clones during the establishment phase. However, over time the mean width of surviving compensated clones will grow, gradually decreasing the likelihood to fluctuate to extinction via random genetic drift.

In our simulations, once a clone grows to a width larger than $w_{escape} = 6$ cells, the probability of genetic drift overcoming inflation becomes smaller than 1%. Clones reaching this escaped phase not only persist virtually indefinitely but continue to grow linearly in lateral width until treatment is initiated at some future time.

A direct consequence of the observed quasi-stability of slower-growing clones is that compensatory mutations can occur over a wide range of population radii. This raises the question of how the probability of a compensated clone to permanently escape selection depends on the radius $r^*$ at which the compensatory mutation occurred and the transition phase is initiated.

In our agent-based simulations we can address this question by triggering compensatory mutations at a specific radius $r^*$ and then measure the effects on subsequent evolutionary dynamics (see Supplementary Movie 2). We find that the rate at which the mean width of compensated clones increases is inversely correlated with $r^*$, consistent with a weaker inflation at higher radii (Fig. 6i). From the slope of the linear width increase we calculate the expected radial expansion $\Delta r_{escape}(r^*)$ that is needed for a compensated subclone to transition to a width $w > w_{escape} = 6$ cells (Fig. 6j). Since $\Delta r_{escape}(r^*)$ increases with $r^*$, compensated clones emerging at larger $r^*$ have to, on average, endure a longer time in the transition phase which results in a decreasing probability $P_{esc}(r^*)$ of the clone to reach escape width (Fig. 6k).

Since $P_{esc}(r^*)$ only describes the dynamics after acquisition of the compensatory mutations, we have to multiply it with the probability $P_{mutate}(r^*)$ of an uncompensated clone to mutate at $r^*$ to obtain the total probability $P_{total}(r^*) = P_{esc}(r^*) \cdot P_{mutate}(r^*)$ that a resistant clone survives due to a compensatory mutation at $r^*$ (Fig. 6k). $P_{mutate}(r) = 1 - (1-\mu)^{w(r)}$ is defined by the width of uncompensated clones and is therefore also subject to the inflation-selection balance. Consequently, $P_{mutate}(r^*)$ initially increases and then saturates while $P_{esc}(r^*)$ continuously decays.

The result is broad peak in the combined probability $P_{total}(r^*)$ suggesting that there is a temporal window during which the evolutionary rescue of a resistant clone is most likely to lead to selection escape, long-term persistence, and eventually treatment failure.

## Discussion

In this work, we study how resistant lineages associated with a fitness cost can be rescued from purifying selection via subsequent compensatory mutations. We introduce an experimental evolution assay based on fluorescence-coupled synthetic mutations in expanding yeast colonies. This model system allows us to track the complete evolutionary trajectories of thousands of individual clones with high spatio-temporal resolution. Three main findings emerged from our study, linking the increased probability of evolutionary rescue and consequences for treatment failure to the growth-induced collective cell dynamics in dense populations.

First, we identify an inflation-selection balance, in which the counteracting effects of peripheral population inflation and selection pressure result in a quasi-stable equilibrium width of slower-growing resistant clones. As a result, these lineages persist at the expanding population front, improving their chance to acquire a subsequent cost-compensatory mutation. Modeling clone boundaries as biased random walkers, we show that a clone-width-dependent effective selection is required for inflation-selection to be stable.

We expect that inflation-selection balance will have a pivotal impact on related processes, such as mutations-selection balance and the maintenance of standing genetic variation, or the accumulation of mutational load and conversional meltdown of populations[21,27–30]

Second, our analysis reveals that a heightened prevalence of compensated resistant clones does not immediately translate to an increased probability of treatment failure but is substantially delayed. De novo compensated clones need to establish and expand above the width of their uncompensated ancestors before having an effect. However, as a result of the stabilizing effect of inflation-selection balance this catching-up takes time. The result is a transient window during which the efficacy of compensatory mutations is negligible. Conceptually, these dynamics have similarities to other types of transients that have been described in range-expanding populations[20].

Third, using an agent-based in silico model of tumor growth, we demonstrate that inflation-selection balance only requires a minimal set of ingredients that is inherent to radially expanding dense populations and also present in solid tumors. We conclude our study by leveraging our simulation platform approach to investigate how the timing of compensatory mutations impacts long-term treatment success.

Our findings suggest a number of future avenues of research. To facilitate a systematic investigation, we focused on a compensation-to-neutral scenario. Using a different wild-type reference strain could extend our analysis to sub-neutral or even net beneficial compensatory mutations. While our well-controlled yeast model system is ideally suited to investigate the fundamental effects of density-driven growth on the evolutionary dynamics of dense populations, it does not capture many of the biological and biochemical intricacies present in other settings, such as active migration, cell-cell adhesion or heterogeneous mechanical properties which may overlay the fundamental processes discussed in this work.

A particularly interesting extension of the work presented here will be to investigate how the secretion of extracellular polymeric substances (EPS), a common feature of bacterial biofilms, might alter the processes discussed in our work[31–33]. While biofilm growth due to EPS secretion, rather than cellular proliferation alone, is likely to result in some form of long-range correlations in cell motion, expansions may not necessarily be lateral. Dedicated experimental assays and tailored simulations will be needed to elucidate the role of EPS and biofilm growth modes on inflation-selection balance and evolutionary rescue dynamics. Moreover, populations growing under spatial constraints, such as confinement in a pore or a tube, might not follow the simple radial inflation discussed here. The ensuing effects on front curvature have the potential to either reduce or amplify local inflation, thereby shifting the point of inflation-selection balance.

In addition, our study focuses on the evolutionary rescue of a pre-existing resistance mutation. Alternative trajectories to two-step resistance, such as the hitchhiking of a resistance mutation on an independent driver mutation, have not been included[34]. However, our experimental strategy and the presented computational framework could be generalized in the future to investigate these alternative scenarios. In experiments, this can be achieved by the implementation of a two-step synthetic mutation platform via two orthogonal recombinase systems.

Notably, our observations might not be exclusive to dense populations. While the necessary ingredients for inflation-selection balance inherently emerge from density-driven growth, they might also be generated by other forms of negative size-dependent selection, such as it has been described for mutualistic scenarios[35].

We predict the existence of an inflation-selection balance to be robust across a wide range of growth parameters and systems. Yet, natural cellular populations exhibit many layers of complexity that are not captured by the minimal assumptions of our model and that might overlay, reshape or even eliminate inflation-selection balance and its effects on evolution. In addition, this study focuses on a limited set of parameters to unravel the fundamental concepts of inflation-selection balance and evolutionary rescue. However, while the investigated fitness costs are within realistic parameter regimes,

we emphasize that they are subject to experimental and computational constraints and do not necessarily reflect the full spectrum of real-world scenarios.

In contrast to reports on the cost of resistance, measurements of the rates of compensatory mutations in natural systems remain elusive. However, it is reasonable to assume that values are likely to be highly variable and potentially outside the range of parameters explored in our experiments[5]. While the concept of inflation-selection balance does not depend on the rate of compensatory mutations, this variability will need to be factored in when interpreting our results in the context of evolutionary rescue in pathogenic biofilms or cancer.

Of particular interest will be to extend our investigations to three-dimensional range expansions, such as matrix-embedded microbial spheroids and cancer cell tumoroids. In addition, including cell death in the context of treatment failure dynamics to study the competitive release of internal resistant cells will be of pivotal importance[36–38].

In conclusion, our work is a crucial step toward a bottom-up understanding of evolutionary dynamics in dense cellular populations as an emergent phenomenon in actively proliferating granular matter. This study is, to our knowledge, the first to combine the effects of two of the most fundamental features of these systems, growth-driven radial expansion and collective dynamics, to systematically investigate their impact on critical evolutionary processes, such as evolutionary rescue and drug resistance evolution.

We expect that future expansions of the framework and tools established here will help to design better quantitative and predictive models for evolution in compact populations, thus serving as a stepping stone toward novel evolution-based therapy strategies[4,39–41].

## Methods

### Strains

All experiments in this work were conducted with the non-motile yeast *S. cerevisiae*. The used strains, yJK26 (resistant mutant with synthetic mutation system) and yMG10 (wild-type background), were constructed on the W303 laboratory strain and are based on the common ancestor yJK19, expressing the $\beta$-estradiol inducible pSCW11-Cre-EBD recombinase construct from pDL12[42].

yJK26 and yMG10 feature the same synthetic mutation cassette $P_{ENO2}$-*loxP-ymCherry-kanMX-loxP-FP2-UBQ-cyh2r*, only differing by the secondary fluorescent protein *FP2* behind the cassette, which for yJK26 is *ymCerulean* and for yMG10 *ymCitrine* (see the following section for details on cloning of the synthetic mutation system). In both cases, the fluorescent protein is coupled via a proteolytically cleavable ubiquitin linker to *cyh2r* (*CYH2Q37E*), conveying resistance to the translational inhibitor Cycloheximide. To construct yJK26, we first introduced the synthetic mutation cassette (yJK20) and subsequently replaced the Nourseothricin resistance marker, originally introduced into yJK19 as selection marker for *cre-EBD* insertion, with a HygR resistance from pAG32 (available from www.addgene.com, #35122). Consequently, yJK26 (and its converted version yJK26c) are constructively resistant against hygromycin B, while the converted version of yMG10 (yMG10c) is used as hygromycin-susceptible wild-type strain in our experiments. All insertions were achieved by amplifying the insert via standard PCR, followed by Lithium Acetate transformation and selection. Correct insertion was verified by cross-junctional colony PCR of positively selected clones. The genotypes of the strains are specified in Supplementary Table 1.

### Construction of synthetic mutation cassette

The synthetic mutation cassette was constructed on the basis of pMEW90 (a kind gift of the lab of Andrew Murray, Harvard)[43]. Initially, all parts were amplified via PCR and assembled via Gibson cloning to yield pMG8. To obtain pJK19, we replaced the *ymCitrine* in pMG8 with *ymCerulean* via Gibson assembly.

### Evolutionary rescue assay

Evolutionary rescue experiments were conducted subjecting yJK26 and yMG10c to spatial competition on 2% agar YPD plates. From the colonies grown from a single cell on the agar plate for 2 days, parts of the single colonies were picked and cultured in a liquid YPD medium overnight. The next day cells were reinoculated into fresh medium and regrown for about 3 h. The two strains were mixed in the ratio 1:9 resistant to susceptible, estimating cell concentration by $OD_{600}$ measurement. 1 µL of mixed cultures of $OD_{600} \approx 20$ were then inoculated on the 6-well plates filled with 15 mL medium and air dried. Image of the colony ~1 h after inoculation shown in Supplementary Fig. 13. Cycloheximide and $\beta$-estradiol were premixed in the medium at desired concentrations (50 nM for cycloheximide, and 2, 4, or 6 nM for estradiol). Colonies were grown for 5 or 9 days prior to treatment and imaged daily. On the treatment day, hygromycin B was applied by pipetting 90 µL of $41.5 \frac{mg}{mL}$ stock as small drops at the edges of the wells. A total of 92 colonies was grown, with at least 5 colonies in the same chemical environment. Treatment after 5 days was initiated for 6 colonies of each condition (50 nM cycloheximide + 0, 4, and 6 nM estradiol and 0 nM cycloheximide + 0 nM estradiol (compensated inoculum)), except 0 nM cycloheximide + 0 nM estradiol (uncompensated inoculum), where only 5 colonies were treated. Treatment on day 9 was done for 18 colonies with 50 nM cycloheximide + 0 and 4 nM estradiol, 17 colonies with 50 nM cycloheximide + 6 nM estradiol, 6 and 5 colonies with no chemicals for uncompensated and compensated inocula correspondingly. The initial number of clones is estimated by manually counting clones from the single-cell resolution images. From 12 colonies we measure the mean of 156 clones per colony, which results in a total of ≈14,500–15,000 clones.

Fitness differences between the strains under different cycloheximide concentrations were measured using the final images of colonies grown from the inoculations with low fractions (2.5–10%) of faster-growing cells (yMG10c) in the slower-growing (yJK26). The opening angles of the faster-growing clones were measured at different radii of the colony and fitted with Eq. 10 from ref. [20] to compute growth rate differences. For 50 nM cycloheximide fitness difference between yJK26 and yJK10c is $s = 1.26 \pm 0.64\%$ (measured using 14 non-interfering clones). For yJK26 and yMG10c under no cycloheximide $s = 0.09 \pm 0.08\%$ (measured using 8 non-interfering clones). The fitness difference between yJK26c and yMG10c is neutral within the errors of the method. Colony examples and measured values are presented in Supplementary Fig. 2a, b.

Mutation rates of yJK26 were estimated by measuring the frequency change of mutated clones throughout colony growth. 1 µL of yJK26 cells in YPD with $OD_{600} \approx 20$ was inoculated onto agar plates containing different concentrations of estradiol and imaged daily. The frequency of compensated mutants at the colony periphery was measured as a function of colony radius and averaged over 6 colonies (see Supplementary Fig. 14) for 0, 2, and 4 nM estradiol and 12 colonies for 6 nM estradiol. Assuming neutrality, any change in mutant frequency only occurs due to mutations and can be described by

$$f_{blue} = 1 - f_{red}e^{\mu(r_0 - r)},$$

with $f_{red}$ being the frequency of red clones, $\mu$ the mutation rate, $r_0$ the initial radius, and $r$ representing the current colony radius. Fitting the function above to the measured frequency change yields the mutation rate per cell per radial colony growth. Orthogonal distance regression was used for fitting and calculating uncertainties of the mutation rate. See Supplementary Fig. 2c, d.

### Imaging and analysis

Colonies were imaged using ZEN 3.0 (blue edition) on a Zeiss Axiozoom V16 fluorescence microscope with PlanNeoFluar Z 2.3x/0.57 objective for imaging right after the inoculation and after 1 day of

growth. A PlanApo Z 0.5x/0.125 objective was used for all other time points. For presentation in Figures, fluorescence images were processes in FIJI (V 2.0.0) to adjust color maps. A custom semi-automated pipeline was used to process the images. First, images were segmented via Ilastik (V 1.3.3), a Machine Learning based segmentation platform[44]. The algorithm was trained on randomly picked images from different days (training data is available upon request). Colony microscopy images with segmented outputs are shown in Supplementary Fig. 15. Segmented images were further analyzed via custom MATLAB (R2020a) and python 3.8 algorithms.

## Clone trajectories
Clone trajectories are reconstructed by assigning labels to the each individual clone at the periphery. For labeling, angular positions throughout different days of the imaging were used. Image after one day of growth was taken as the reference (Fig. 2b). For each day clone angular positions were compared to clone positions one day before to assign IDs. Full trajectory tracking was only done for illustrative purposes for one colony and was not needed in the quantitative analysis presented in the main text.

## Categorization of mixed clones
The transition from uncompensated to fully compensated states in both experiments and simulations typically progresses via a mixed state in which the clone is comprised of both compensated and uncompensated cells at the front. Since experimental data is subject to imaging constraints and segmentation artifacts, we categorized clones with any detectable compensation as fully compensated, reflected in the apparently sharp transitions in Fig. 2b. However, we excluded any such ambiguous regions for the width analysis in Fig. 2c to avoid systematic bias due to segmentation artifacts. These limitations due not apply to the categorization of resurgent growth domes so that the treatment failure analysis in Fig. 3 includes all compensated clones even if some uncompensated cells remain (see Supplementary Fig. 6). For both radial-random-walk and agent-based simulations, featuring perfect type resolution, uncompensated and compensated subclones were considered independently in the width analysis (Figs. 5c, 6c). Note that in agent-based-simulations uncompensated cells in the population bulk can still mutate, resulting in an apparent intermixing (see Supplementary Video 2). However, the effect of these bulk mutations on the evolutionary dynamics at the front and the fate of the clone is negligible.

## Estimation of width distributions
Clone width distributions (panels d and e of Figs. 2, 5, and 6) represent smoothed values with log-transformed kernel density estimation (KDE)[45]. Supplementary Fig. 16 shows the comparison of the underlying histogram to the probability density estimations using either an asymmetrical gamma function kernel[46], a conventional Gaussian kernel, or a log-transformed Gaussian kernel. The advantage of a log-transformed KDE is that it produces the estimate for strictly positive observations and is zero at $x = 0$ unlike conventional Gaussian KDE or standard KDE with asymmetrical kernels[45]. For the experimental data (Fig. 2) we apply a cut-off of one-pixel size to avoid segmentation-associated artifacts.

## Description of the random walker model
Random walker simulations were implemented in python 3.8. We treat the boundaries between each resistant clone with the wild type as independent non-interacting one-dimensional random walkers in angular space[19,23]:

$$\Phi[i] = \Phi[i-1] + \eta(b, \sigma_X)\frac{1}{R}.$$

Where $\eta(\mu, \sigma)$ is normally distributed noise with mean $\mu$ and variance $\sigma$. $b$ is the bias of the walk, which is zero for the neutral clones, and can be positive or negative number for clones with a fitness deficit, depending on the side of the sector that the random walker is representing. Each boundary performs a random walk, that is a solution to the diffusion equation. The diffusion coefficient is a measure of the genetic drift in the system. It is represented by the random step that follows Gaussian distribution with the variance[19]:

$$<\Delta X^2> = \sigma_X^2 = 4D.$$

To simulate clonal extinction we cause neighboring random walkers to be annihilated when they cross. The bias of the random walk $b$ is calculated using the equal-time argument in refs. [19,20]:

$$b = \sqrt{s(2+s)}.$$

To test our hypothesis of inflation-selection balance emerging from collective cell dynamics we use an effective width-dependent selection coefficient $s = s(w)$, with a shape similar to that experimentally observed in Fig. 2g of ref. [13]. Here we used

$$s_{\text{eff}} = s_0 \frac{2}{1 + e^{\frac{w_c}{w}}},$$

where $w_c$ is the critical width analogous to the critical width in ref. [13] and is a free parameter.

## Parameterization of the random walker model
To parameterize the random walk model, we use the measured initial radius of the colony after inoculation and the measured fitness cost $s_0$ (see Supplementary Table 2). To obtain the value for $w_c$ in the effective selection we compared the equilibrium width of slower-growing clones to the experimentally measured one. For a fitness cost of $s_0 = 0.013$ a critical width of $w_c = 280$ μm best matched the experimental data. To estimate genetic drift in the system, we tuned the diffusion coefficient together with initial clone size distribution to match the experimental clone survival probability (Supplementary Figs. 17 and 18). The diffusion coefficient to best match experiments is 0.23 μm⁻¹. The strength of genetic drift controls the shift of survival probability. The initial width distribution controls the slope of the initial drop in survival probability, when most of the clones are of very small size. Here, our initial size distribution that followed a Gaussian distribution with the mean of 20 μm and standard deviation of 5 μm. For comparison experimental mean width of red clones after 1 day is $26 \pm 12$ μm. Analogously, we compared the results of different mutation rates (see Supplementary Fig. 19). Mutation rate used in the simulations $\mu = 10^{-4}$ μm⁻¹ is in a good agreement with experimentally measured value of $\mu = 2.65 \pm 0.25 \times 10^{-4}$ μm⁻¹.

## Agent-based simulations
Agend-based simulations of in silico tumors were performed via a modified version of PhysiCell (version 1.8.0), an open-source platform for 2D and 3D tumor growth[26], with BioFVM (version 1.1.6)[47] to solve the transport equation.

The model simulates growth, including mechanical cell-cell interactions, within a chemical microenvironment that is modeled via diffusion of chemicals in the system. Cell division and physical forces exerted on the neighbors lead to densely packed population. Agents continuously consume nutrients diffusing to the system from the outer edge with circular Dirichlet boundary conditions. This results in the emergence of a growth layer as nutrients become depleted in the population bulk (see Supplementary Movie 3).

Each cell in the simulation carries its own characteristics, defined by its mechanical and chemical environment. These characteristics

include cell size, cycle progression, growth rate, and mechanics. We initialize the system similar to the initial inoculations in experiments, having a hollow dense ring of cells, with sufficiently spaced single slower-growing mutants interspersed at the periphery (16.6%). Growth of both cell types depends on the nutrient concentration, slowing down gradually until the growth is completely halted for very low concentrations. For parameterization of the simulation see Supplementary Table 3.

The ratio of cell size to the colony radius was chosen to differ from the experiment to accommodate computational cost limitations. Virtual populations are grown up to $\sim 2 \times 10^6$ cells, corresponding to a 10-fold radial increase from the starting colony.

### Reporting summary

Further information on research design is available in the Nature Portfolio Reporting Summary linked to this article.

## Data availability

Imaging data used in this study is available at https://figshare.com/projects/Aif2022_NatComms/146175. The Source data for figures are provided together with the code to reproduce the figures at https://gitlab.gwdg.de/kayser-lab/aif_isb. See README.md file for detailed description. Source data for Fig. 5 and random walk simulations related Supplementary Figures are available from the authors upon request due to very large data size. These data sets can also be obtained by rerunning simulations provided in the repository as described in the Fig. 5 section of the README.md. Source data are provided with this paper.

## Code availability

The code used for simulations and analysis in this study is available at https://gitlab.gwdg.de/kayser-lab/aif_isb. Persitent Identifier for the code 21.11101/0000-0007-F9A6-5 can be resolved at https://www.pidconsortium.net/.

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

## Acknowledgements

This work was supported by the Emmy Noether Programme of the German Research Foundation (project 455449456) and the National Institute of General Medical Sciences of the National Institutes of Health under award 2R01GM115851-06A1. J.K. acknowledges a research scholarship (KA 4486/1-1) awarded by the German Research Foundation. O.H. acknowledges support by a Humboldt Professorship of the Alexander von Humboldt Foundation. The authors thank M. Eiche, C. Moeckel, A. Feder, B. Good, M. Gralka, C.F. Schreck, and D. Fusco for vital discussions, as well as J. Guck and his group for their invaluable support. The help of the groups of J. Rine and A. Murray, providing critical genetic material and invaluable insights into yeast genetics, is gratefully acknowledged.

## Author contributions

S.A., O.H., and J.K. conceived and designed the study. S.A., N.A., L.K., and J.K. carried out and analyzed the experiments. S.A. and J.K. performed and evaluated the simulations. S.A., N.A., O.H., and J.K. discussed and interpreted the results. S.A. and J.K. wrote the manuscript.

## Funding

## Competing interests

The authors declare no competing interests.
