## [Peer Review File · Nature Communications]

Evolutionary rescue of resistant mutants is governed by a balance between radial expansion and selection in compact populationsReviewers' Comments:

Reviewer #1:

Remarks to the Author:

The authors tackle an important and long-standing question: How can cells with a drug resistance mutation remain in the population when the drug is not provided, given that the resistance mutations usually incur a fitness cost? The authors experimentally study this question by focusing on a colony growth model system, which involves densely packed yeast cells with spatial structure. In addition, the authors use agent-based simulations to generalize their results and interpretations from the experimental system.

The authors claim the following key advances:

- The authors introduce a new experimental system that enables them to identify compensatory mutations by a fluorescence signal, which allows them to track the emergence and fate of mutations in space and time using fluorescence microscopy.
- Drug resistant strains have a lower growth rate so that they should be (quickly) lost from the frontier of expanding populations. However, the authors find that the drug resistant clones prevail in the expanding colony much longer than predicted by a null-model of radial range expansion with a selection biased random walk. Because the drug-resistant clones prevail for so long, the absolute probability of acquiring a compensatory mutation to offset the growth rate deficit is increased.
- The authors discover that the surprisingly long prevalence of uncompensated clones in the expanding population results from a stable equilibrium between the effects of radial expansion and selection (which they term "inflation-selection balance"). The authors previously showed that the key ingredient to this interpretation, the clone-width-dependent selection, can arise due to mechanical coupling of cells in dense expanding colonies.
- The interpretations are supported by agent-based simulations inspired by tumor growth. These simulations also show that the experimental findings are not yeast-specific.

In my opinion the data presented by the authors clearly supports these claimed advances. The fundamental (i.e. system-independent) nature of the stable equilibrium between radial expansion and selection to stabilize uncompensated drug-resistant clones makes this manuscript an important advance for the field.

The supporting information comprehensively covers control experiments/simulations.

Below, I am listing a few major and minor weaknesses of the manuscript:

Major:

1. A weakness of the study is that the authors study a synthetic system in which they artificially decreased the growth rate of the drug-resistant cells by adding another drug (cycloheximide), to which the WT is resistant. Similarly, the mutation rate is controlled by varying concentrations of beta-estradiol. Although the system enables exquisite experimental control, the system is quite artificial and in a parameter regime that may not reflect realistic conditions in drug resistance evolution. Can the authors show that the inflation-selection balance also occurs in "realistic" parameter regimes?
2. Important, but easy to fix: Throughout the manuscript, the authors often refer to "less-fit resistant clones". It is not immediately clear to me what the fitness currency is – probably the cellular growth rate. Please clarify the fitness currency. If it is the growth rate, I suggest to use the terminology of lower/higher growth rate instead of less/increased fitness.

Minor:

3. The Introduction is balanced and an appropriate summary of the state of the art. However, I suggest to break this into at least two paragraphs to improve the readability.
4. The font sizes in all figures are very small. In Figure 6 the font sizes are tiny. Please increase.
5. Fig. 1F: I am not sure what the shading of the genes and promoter regions is supposed to indicate.

If it has no meaning, please remove the shading.

6. Line 167: grammatical error.

7. In the main text, the values for the radius r are sometimes given without a unit. See line 188, line 222, line 226. Can the authors please check all units in their text?

8. Section starting at line 285: The section starts with "we argue that". However, this is a results section. Consider rephrasing as follows: "Motivated by XYZ, we tested the hypothesis that ABC".

9. The manuscript title is perhaps a bit cryptic for a general audience – I am not sure that the jargon "inflation-selection balance" should be used in the title.

Reviewer #2:

Remarks to the Author:

This study adds to a substantial body of recent literature on evolution in expanding populations, especially building on results of Kayser et al. 2019 and Giometto et al. 2018 (refs 13 and 14). The most novel features are the focus on evolutionary rescue, the concept of inflation-selection balance, and the experimental system that permits tracking of drug-resistant lineages with predetermined fitness values. The analytical and computational methods are well chosen, albeit less mathematically sophisticated than much of the previous work in this subfield. Assessing the validity of the experimental methods is beyond my expertise. Overall the results are convincing, well presented, and potentially of wide interest.

Main comments:

The claim that "selection is typically thought to only depend on the relative fitness difference" (line 300) is doubtful in light of refs 13 and 14. As stated in the Introduction, "We and others have recently demonstrated that in dense populations collective cell dynamics inherently decrease the power of selection by several orders of magnitude". Ref 13 concluded that "Potential implications [include] an increased chance of evolutionary rescue". Unless "typically thought" can be supported by citations, I suggest rephrasing the claim and, in subsequent sentences, clarifying that the detection of width-dependent selection isn't new. Likewise, the "Intriguingly" on line 343 seems uncalled for, given that the current study can be seen as a continuation of ref 13. The new work is interesting enough that the authors can afford to avoid any suspicion of overselling novelty.

The "Agent based simulation animation with random mutations" video shows blue and red cells intermixed but in the experimental system (Figs 1g, 2b, S6b, S14) the two types appear to be separated by sharp boundaries. Why the difference? Is it real or illusory? Does it matter? How can the intermixing seen in simulations be reconciled with the sharp transition model of Fig 6h? On what basis were mottled lineages, such as those seen in the video, categorised as either compensated (blue) or uncompensated (red)? Please answer these questions in the Methods and Results or at least acknowledge that the patterns differ.

The SI says "Code is available on gitLab" but fails to provide a URL. All code should be made public.

Minor comments:

Fig 2f: What's the difference between the green and grey dotted lines?

Fig 5f took me a while to parse because the dark grey and dark green are so hard to distinguish. In particular, two curves (the neutral simulations) mostly overlap, resulting in a dark grey curve with dark green fringes that looks almost exactly the same as the dark grey-green curve representing the width-dependent case with rescue. I suggest using more distinct colours in Figs 1f, 5f, 6f, etc.

Figs 5d, 5e, 6d and 6e seem to show high probabilities of clones having zero width, which is

inconsistent with survival. How exactly were these probability densities calculated? Are high densities at zero a side effect of smoothing? Either find a way to avoid this apparent artefact or explain, in the figure captions and Methods, how to interpret these probability densities.

Fig 6i: It's unclear what this figure is meant to show as the four curves aren't labelled. Please add a legend (or at least an explanation in the caption).

Line 179: It would be helpful to specify here that the "minimal null model" is a computational model / simulation (rather than an experimental or analytical model).

Line 391: "Fig. 2h" should be "Fig. 6h".

Fig S11: The caption says, "Plus and minus indicate areas of growing and shrinking clones" but I don't see any such symbols in the figure.

As shown in Fig 6h and elsewhere, the clones under investigation typically aren't cone shaped, so it's unclear why the paper should repeatedly use the term "cone" to describe them. Couldn't all these "cones" be referred to as clones or lineages? If not then explain the difference. In any case, it'd be worth checking that no "cone" is a typo (see especially line 165).

Although the article is very well written, there are a few typos and minor grammatical mistakes. For example, "a compensatory mutation rates" (line 166); "level of at" (203); "extend of" (242); "stop to proliferate" (370).

Reviewer #3:

Remarks to the Author:

This manuscript presents a synthetic experimental model system for studying evolutionary rescue: the phenomenon where a previously less fit mutant gains an additional mutation that restores its fitness. A "resistant" strain of the yeast *S. cerevisiae* is engineered to be resistant to the antibiotic hygromycin B, but sensitive to the antibiotic cycloheximide (to which the parent strain is resistant), allowing the fitness of the resistant strain to be manipulated via addition of cycloheximide. The resistant strain is further engineered to randomly undergo a genetic switch to a state that is resistant to cycloheximide. The rate of this switch can be controlled by adding beta-estradiol.

Using this system, the authors investigate the dynamics of evolutionary rescue in spatially structured expanding colonies. Their key result relates to the balance between expansion of a clone due to the radial expansion of the growing front, and clone width-dependent selection, which they term "inflation-selection balance".

I find this an impressive piece of work which reveals interesting insights into the effects of population spatial structure on evolution. My comments relate mainly to the interpretation of the results in terms of clinical relevance, and the clarity of some parts of the manuscript.

1. The comparison to the "null model" in lines 177-182 is unclear. It is not explained at this stage what the null model is, or why it is called "null". I guess it is the version of the diffusing-boundaries model without width dependent selection. But the reference is to figure 5, which actually shows a different version of the model, which does include width-dependent selection and agrees well with the experiments. This part should be made much clearer by actually explaining what the model is when it is introduced.

2. Frequently units are missing for r in the text (eg line 188 but also in many other places).

3. In general the first results section would be clarified by separating the results for the compensated and uncompensated clones. For example in lines 195-203 it is not clear what type of clone is being

discussed.

4. It should be clarified that "treatment failure" here refers to a specific context. Here for example there is no killing by the hygromycin B. The results would be different for an antibiotic that killed the bulk of the colony, freeing up space.

5. A claim is made that the results are very general, eg in lines 355-357 they are connected to bacterial biofilms and solid tumours. However the key findings depend crucially on two factors: radial expansion and width-dependent selection, that may not be relevant in bacterial biofilms. Bacterial biofilms do not necessarily expand laterally as they grow. Moreover the surfing mechanism that leads to width-dependent selection might not happen in biofilms where cells are quite sparse and surrounded by exopolysaccharide. I see this work as a very nice model study connecting biophysics, spatial structure and evolution, but I think the claims on general relevance should be toned down.

6. I found the text in lines 410-419, 439-448 and also 451-464 convoluted and hard to follow. These concepts could be explained more clearly.

Response to the Reviewers

Aif et al.

We thank the reviewers for their positive and constructive comments. These have helped us to improve the manuscript, which we have revised to implement the suggested changes. Below we give a detailed point-by-point response to each of the reviewers comments. We hope that the reviewers find their comments adequately addressed and that the revised manuscript can now be accepted for publication.

Reviewer #1 (Remarks to the Author):

The authors tackle an important and long-standing question: How can cells with a drug resistance mutation remain in the population when the drug is not provided, given that the resistance mutations usually incur a fitness cost? The authors experimentally study this question by focusing on a colony growth model system, which involves densely packed yeast cells with spatial structure. In addition, the authors use agent-based simulations to generalize their results and interpretations from the experimental system.

The authors claim the following key advances:

- The authors introduce a new experimental system that enables them to identify compensatory mutations by a fluorescence signal, which allows them to track the emergence and fate of mutations in space and time using fluorescence microscopy.
- Drug resistant strains have a lower growth rate so that they should be (quickly) lost from the frontier of expanding populations. However, the authors find that the drug resistant clones prevail in the expanding colony much longer than predicted by a null-model of radial range expansion with a selection biased random walk. Because the drug-resistant clones prevail for so long, the absolute probability of acquiring a compensatory mutation to offset the growth rate deficit is increased.
- The authors discover that the surprisingly long prevalence of uncompensated clones in the expanding population results from a stable equilibrium between the effects of radial expansion and selection (which they term “inflation-selection balance”). The authors previously showed that the key ingredient to this interpretation, the clone-width-dependent selection, can arise due to mechanical coupling of cells in dense expanding colonies.
- The interpretations are supported by agent-based simulations inspired by tumor growth. These simulations also show that the experimental findings are not yeast-specific.

In my opinion the data presented by the authors clearly supports these claimed advances. The fundamental (i.e. system-independent) nature of the stable equilibrium between radial expansion and selection to stabilize uncompensated drug-resistant clones makes this manuscript an important advance for the field.

The supporting information comprehensively covers control experiments/simulations.

Below, I am listing a few major and minor weaknesses of the manuscript:

We thank the reviewer for their encouraging comments and excellent suggestions, which we included into the revised version of the manuscript. The detailed changes are discussed below.

Major:

Comment 1A

1. A weakness of the study is that the authors study a synthetic system in which they artificially decreased the growth rate of the drug-resistant cells by adding another drug (cycloheximide), to which the WT is resistant. Similarly, the mutation rate is controlled by varying concentrations of beta-estradiol. Although the system enables exquisite

experimental control, the system is quite artificial and in a parameter regime that may not reflect realistic conditions in drug resistance evolution. Can the authors show that the inflation-selection balance also occurs in “realistic” parameter regimes?

We completely agree with the reviewer that the artificial nature of our experimental model system has to be considered when interpreting our results in the context of natural populations. In this study, our primary aim was to unravel the fundamental mechanisms that drive inflation-selection balance and evolutionary rescue, a task for which a well-controlled model system is ideally suited. Yet, we had also carefully chosen the fitness cost in our assays to reflect a realistic parameter regime while accommodating experimental constraints. However, the reviewer is correct in pointing out that we had not clearly discussed the chosen set of parameters in the context of the range reported for natural populations.

Resistance-associated growth rate reductions have been reported to vary substantially depending on the type of cell, the drug and the mechanism of resistance. For bacteria, changes of doubling times from 0 to 400% (equivalent to a fitness cost of $s = 0 - 0.75$) have been observed *in vitro* (see table 1 in ref. [2]). However, data from clinical isolates suggest that low-cost variants are most frequent and typically do not exceed a cost of $s = 0.3$ (see ref. [22]). Data on the cost of resistance in cancer is more scarce but reported values reach from $s = 0 - 0.33$ (ref. [37]).

We chose the moderately low fitness cost of $s = 0.013$ (induced by 50 nM cycloheximide) to both reflect this real-world distribution of fitness effects while also facilitating optimal data acquisition within the constraints of our experimental assay, such as the minimum size of the initial inoculum or the maximum attainable colony size.. Our agent-based simulations allow us to extend the regime to smaller initial population sizes (and therefore stronger inflation), for which we increased fitness costs to up to $s = 0.21$. To further test the robustness, we conducted additional simulations with $s = 0.16$ and $s = 0.11$, now included in the supplementary information (Fig. S12). We observed inflation-selection balance in all simulations and experiments, corroborating our notion that inflation-selection balance indeed can occur throughout the “realistic” regime of resistance-associated fitness costs.

Measurements of the rates of spontaneously occurring compensatory mutations remain much more elusive. Yet, it is reasonable to assume that compensatory mutation rates in natural systems are likely to be highly variable and may potentially be outside the range of parameters explored in this work (see ref. [5]). In our experiments and simulations, we optimized the mutation rate to obtain a sufficient number of evolutionary rescue events within the accessible time frame while maintaining a low probability of clonal interference of multiple compensated subclones within one resistant lineage. This allowed us to study in great detail how evolutionary rescue dynamics are shifted by inflation-selection balance. However, the resulting absolute values for the probability of evolutionary rescue and treatment failure, such as those shown in Fig. 3a,b or Fig. 6k, are certainly subject to variations in compensatory mutation rate and may differ from those in natural populations.

Finally, we acknowledge that the inherently artificial nature of our model system will remain even for the most carefully chosen set of parameters. Future work will therefore have to explore the roles of other system characteristics, such as different growth modes or population geometries, to gradually approach the rich complexity of natural populations.

To improve our manuscript, we now discuss the chosen set of parameters and the underlying rationale when first introducing the synthetic mutation system. In particular, we now include a more thorough comparison to the range of values reported for natural populations in the literature.

We also conducted additional simulations with varying magnitudes of resistance-associated fitness costs and included the results in the supplementary information. Moreover, we extended our discussion to clarify the limitations of our approach, including the artificial nature of our model system.

List of Changes:

- When first introducing our model system, we now discuss the resistance-associated fitness costs in our study in the context of realistic regimes (lines 119-128)
- We improved the section discussing the rationale underlying our choice of parameters in the context of experimental and computational limitations (line 635 ff)
- We added Supplementary Figure S 12 showing additional simulation results for different fitness cost values.

- We included a new paragraph in our discussion to elaborate on the choice of compensatory mutation rate and ensuing limitations for the interpretation of our results (line 648 ff).

Comment 1B

2. Important, but easy to fix: Throughout the manuscript, the authors often refer to “less-fit resistant clones”. It is not immediately clear to me what the fitness currency is – probably the cellular growth rate. Please clarify the fitness currency. If it is the growth rate, I suggest to use the terminology of lower/higher growth rate instead of less/increased fitness.

We thank the reviewer for pointing out this important aspect which, we agree, needs further clarification. Qualitatively, less-fit clones are indeed comprised of slower-growing cells and we now indicate the link between a decreased growth rate and the term “fitness cost of resistance” in the beginning of the introduction. However, it should be noted that in our spatially-structured, dense populations quantitative measures of relative fitness cost of a clone refer to the rate of change in clone width, rather than the doubling rate of the comprising cells. We now indicate this currency of fitness more clearly in the main text.

List of Changes:

- Line 40, the initial sentence of the introduction, now states: *“Many drug resistance mutations are associated with a decrease in growth rate in the absence of treatment, a phenomenon often referred to as the fitness cost of resistance, and thus subject to purifying selection”*
- Line 48 (the first mentioning of “less-fit”) now states: *“However, the short lifetime and small size of less-fit intermediate clones, comprised of slower-growing cells, makes crossing such a “fitness valley” inherently rare, requiring large populations”*
- Line 132 in the results section now reads *“In addition, it should be noted that quantitative measures of the relative fitness cost refer to the rate of change in clone width, rather than the doubling rate of the comprising cells”*

Minor:

Comment 1C

3. The Introduction is balanced and an appropriate summary of the state of the art. However, I suggest to break this into at least two paragraphs to improve the readability.

We completely agree with the reviewer.

List of Changes:

- We divided the introduction into three paragraphs to increase readability.

Comment 1D

4. The font sizes in all figures are very small. In Figure 6 the font sizes are tiny. Please increase.

Thank you. We increased the font sizes.

List of Changes:

- We increased the font sizes from 5-6 to 7-8 pt.

Comment 1E

5. Fig. 1F: I am not sure what the shading of the genes and promoter regions is supposed to indicate. If it has no meaning, please remove the shading.

The shading of the genes indeed did not have a specific meaning. We removed it.

List of Changes:

- We removed the shading of the genes and promoter regions in Figure 1f, as it did not provide any additional information.

Comment 1F

6. Line 167: grammatical error.

Thank you.

List of Changes:

- We corrected the error. The sentence now reads *“Clones were initially associated with an intrinsic fitness cost of $s = 0.013 \pm 0.006$ but could acquire a compensatory mutation at rate μ (see figure captions).”*

Comment 1G

7. In the main text, the values for the radius r are sometimes given without a unit. See line 188, line 222, line 226. Can the authors please check all units in their text?

Thank you for pointing out this omission.

List of Changes:

- We included units for all values of r in the main text.

Comment 1H

8. Section starting at line 285: The section starts with “we argue that”. However, this is a results section. Consider rephrasing as follows: “Motivated by XYZ, we tested the hypothesis that ABC”.

This is an excellent suggestion. We changed the text accordingly.

List of Changes:

- The passage now reads *“Motivated by the observed width stability of uncompensated clones, we tested our hypothesis that this phenomenon might also be a key determinant of evolutionary rescue dynamics, including the window of inefficacy. The origin of the plateau in width...”*

Comment 1I

9. The manuscript title is perhaps a bit cryptic for a general audience – I am not sure that the jargon “inflation-selection balance” should be used in the title.

The reviewer does have a point. While inflation-selection balance is a central pillar of this work, the term has not yet been coined. We cleared our title of this jargon and replaced it with a more descriptive version.

List of changes:

- We changed the title to “Evolutionary rescue of resistant mutants is governed by a balance between radial expansion and selection in compact populations”

Reviewer #2 (Remarks to the Author):

This study adds to a substantial body of recent literature on evolution in expanding populations, especially building on results of Kayser et al. 2019 and Giometto et al. 2018 (refs 13 and 14). The most novel features are the focus on evolutionary rescue, the concept of inflation-selection balance, and the experimental system that permits tracking of drug-resistant lineages with predetermined fitness values. The analytical and computational methods are well chosen, albeit less mathematically sophisticated than much of the previous work in this subfield. Assessing the validity of the experimental methods is beyond my expertise. Overall the results are convincing, well presented, and potentially of wide interest.

We thank the reviewer for their encouraging assessment of our work and the valuable suggestions on how to further improve our manuscript. We incorporated the suggested changes, all of which we agree with, into the revised version and discuss all changes in detail below.

Main comments:

Comment 2A

The claim that "selection is typically thought to only depend on the relative fitness difference" (line 300) is doubtful in light of refs 13 and 14. As stated in the Introduction, "We and others have recently demonstrated that in dense populations collective cell dynamics inherently decrease the power of selection by several orders of magnitude". Ref 13 concluded that "Potential implications [include] an increased chance of evolutionary rescue". Unless "typically thought" can be supported by citations, I suggest rephrasing the claim and, in subsequent sentences, clarifying that the detection of width-dependent selection isn't new. Likewise, the "Intriguingly" on line 343 seems uncalled for, given that the current study can be seen as a continuation of ref 13. The new work is interesting enough that the authors can afford to avoid any suspicion of overselling novelty.

We agree that this is an important point and thank the reviewer for encouraging us to differentiate more clearly between the new insights described in this manuscript and those obtained in our previous work. We rephrased the indicated section per the reviewer's suggestion. In particular, we now introduce the "constant selection" scenario as the *simplest*, rather than the *typical* scenario, while also citing important work studying this scenario. In addition, we now point out more clearly that we and others had already demonstrated width-dependent selection in previous work. We also agree that the "Intriguingly" might be perceived as overselling the novelty of our findings and changed the wording accordingly.

List of Changes:

- Line 342 ff now reads "*In the simplest scenario, inflation depends on the sector width and the current population radius as $\partial r w_{inf}(w, r) = w/r$ while selection only depend on the relative fitness difference s of the adjacent clones $\partial r w_{sel} = \sqrt{|s|(2 + s)}$. Such a "constant selection" scenario has been comprehensively investigated in previous studies of range expansions [18–20, 22].*"
- Line 356 ff now reads "*However, we and others have recently demonstrated that in dense populations, similar to those investigated here, selection decreases with clone width [13, 14]. For such a width-dependent selection $\partial r w_{sel}(w)$, a stable equilibrium width can exist, such that small clones are inflation dominated while bigger clones are selection dominated (Fig. 4d).*"
- We deleted the "Intriguingly" in line 343 of the initial manuscript such that line 392 now simply states "*We previously demonstrated that...*"

Comment 2B

The "Agent based simulation animation with random mutations" video shows blue and red cells intermixed but in the experimental system (Figs 1g, 2b, S6b, S14) the two types appear to be separated by sharp boundaries. Why the difference? Is it real or illusory? Does it matter? How can the intermixing seen in simulations be reconciled with the sharp transition model of Fig 6h? On what basis were mottled lineages, such as those seen in the video, categorised as either compensated (blue) or uncompensated (red)? Please answer these questions in the Methods and Results or at least acknowledge that the patterns differ.

This is a keen and important observation! We thank the reviewer to point out this potential point of confusion and agree that a more detailed explanation is warranted.

The transition from uncompensated to fully compensated states in both experiments and simulations almost always progresses via an mixed state in which the clone is comprised of both compensated and uncompensated cells at the front. The only exception is that of a single-cell-wide uncompensated clone that completely switches to full compensation by a single mutation event. The clone displayed in the former Fig. 6h was an unfortunate choice, as it displayed exactly such a rare event. We now updated Fig. 6h to show a more representative scenario.

In addition to partially compensated clones at the front, uncompensated cells in the population bulk can still mutate, exacerbating the appearance of intermixing. These bulk mutants are the main source of perceived intermixing in the agent-based simulation animation with random mutations. However, even though these mutation events are real, their effect on the fate of the clone is negligible. Disentangling the evolutionary dynamics at the front from those in the bulk was the primary reason why we conducted time-resolved experiments instead of purely relying on end-point measurements of colony composition.

How we categorized and displayed different states of compensation at the front depends on the type of underlying data. In brief, experimental data is subject to imaging constraints and segmentation artifacts. We therefore categorized clones with any detectable compensation as fully compensated, reflected in the apparently sharp transitions in Fig. 2b. However, in reality these transitions will also be gradual. In fact, upon close inspection of Fig. 1g one can still discern red uncompensated cells alongside the nascent blue compensated cells right after the rescue event.

Simulations, in contrast, have a perfect resolution without the need for segmentation and we therefore display the full detailed composition information in Figs. 5b and 6b.

We acknowledge the complexity of the scenarios and processes outlined above and added a dedicated section "Categorization of mixed clones" to our Methods to clarify our analysis.

List of Changes:

- We include a new subsection in Methods "Categorization of mixed clones", where we explain the difference in trajectory patterns between experiments and agent-based simulations.
- We updated Figure 6h to display the typical scenario of a clone with transient mixed state.

Comment 2C

The SI says "Code is available on gitLab" but fails to provide a URL. All code should be made public.

Thank you for pointing out this omission. We made all our code public:

https://gitlab.gwdg.de/kayser-lab/aif_isb

List of Changes:

- We added Data and Code availability statements and our code has been made public. It can be accessed via the gitlab link: https://gitlab.gwdg.de/kayser-lab/aif_isb

Minor comments:

Comment 2D

Fig 2f: What's the difference between the green and grey dotted lines?

The green dotted line represent data from null-model-simulations *with* mutations while the grey dotted line show the data of the *no-mutations* control. We expanded our legend for clarification.

List of Changes:

- We improved the legend.

Comment 2E

Fig 5f took me a while to parse because the dark grey and dark green are so hard to distinguish. In particular, two curves (the neutral simulations) mostly overlap, resulting in a dark grey curve with dark green fringes that looks almost exactly the same as the dark grey-green curve representing the width-dependent case with rescue. I suggest using more distinct colours in Figs 1f, 5f, 6f, etc.a

Thank you, we adapted the colour palette for enhanced contrast and added a clarifying sentence to the caption of Fig. 5f.

List of Changes:

- We changed green colours to a lighter hue for better contrast between samples.
- We added the following sentence to the caption of Fig. 5f: “*Note that with-rescue and no-rescue data overlaps for the neutral scenario.*”

Comment 2E

Figs 5d, 5e, 6d and 6e seem to show high probabilities of clones having zero width, which is inconsistent with survival. How exactly were these probability densities calculated? Are high densities at zero a side effect of smoothing? Either find a way to avoid this apparent artefact or explain, in the figure captions and Methods, how to interpret these probability densities.

We thank the reviewer for the keen observation and pointing out this inconsistency. As the reviewer already suspected, the non-zero probability for clones of size zero are a smoothing artefact. To avoid binning artefacts, we opted for a kernel density estimation approach to calculate the probability densities. However, the previously employed symmetric Gaussian kernel had the disadvantage of generating apparent non-zero values extending to the negative width axis. This effect was further exacerbated by single-pixel-level segmentation artefacts.

We now solved this issue by employing an asymmetric kernel (see new reference [45]) and applying a single-pixel cut-off for imaging-based data to avoid segmentation-associated artefacts. Note, that the trade-off for using the new log-transformed kernel is the potential for non-monotonic behavior of the PDF close to zero-width. However, this does not influence the overall shape of the PDF and we therefore deem the new approach superior to the one used previously.

To exemplify these different types of PDF calculation, including potential artefacts, and their relation to the underlying histogram, we included a more detailed explanation in the methods section that is further supported by accompanying figures in the supplementary material (see new Supplementary Fig. S16).

List of Changes:

- We implemented a log-transformed kernel density estimation for the probability density calculations as in reference [45] ;
- We implemented a one-pixel cut-off for the width analysis of experimental data to avoid pixel-level segmentation artefacts.
- Figs. 5d, 5e, 6d and 6e now show the updated data
- We included a detailed description of our procedure in the new dedicated “Estimation of width distributions” section in the Methods
- We included a new supplementary Figure, comparing symmetric and asymmetric kernels in the supplementary information (new supplementary Fig. S16)

Comment 2G

Fig 6i: It's unclear what this figure is meant to show as the four curves aren't labelled. Please add a legend (or at least an explanation in the caption).

Thank you for pointing out this critical omission. We added labels to Fig. 6i and explained in the caption what the different curves mean.

List of Changes:

- We added the missing labels to Fig 6i
- We added an explanation to Figure 6i in the caption.

Comment 2H

Line 179: It would be helpful to specify here that the "minimal null model" is a computational model / simulation (rather than an experimental or analytical model).

This is an excellent suggestion.

List of Changes:

- In lines 206 ff we now state: "We then compared our experimental results to a baseline obtained from a minimal *in silico* null model of radial range expansion. In this model, the trajectories of individual sector boundaries are simulated as selection-biased 1D random walks on a radially expanding periphery..."

Comment 2I

Line 391: "Fig. 2h" should be "Fig. 6h".

Thank you.

List of Changes:

- We now refer to Fig. 6h instead of Fig. 2h

Comment 2J

Fig S11: The caption says, "Plus and minus indicate areas of growing and shrinking clones" but I don't see any such symbols in the figure.

Thank you. We added the symbols.

List of Changes:

- Added + and - mentioned in the caption to the figure.

Comment 2K

As shown in Fig 6h and elsewhere, the clones under investigation typically aren't cone shaped, so it's unclear why the paper should repeatedly use the term "cone" to describe them. Couldn't all these "cones" be referred to as clones or lineages? If not then explain the difference. In any case, it'd be worth checking that no "cone" is a typo (see especially line 165).

We thank the reviewer for this great suggestion on how to improve the clarity of our text. In the original manuscript, we used the term *cone*, referring to the semi-circular resurgent growth pattern that originates from small resistant clones after drug application, to differentiate them from the streak- or sector-like geometry prior to treatment onset. We agree that the similarity of these terms might lead to unnecessary confusion, especially given that the resurgent growth patterns presented in this work are rather dome shaped than cone shaped.

List of Changes:

- Line 139 ff now introduces the term "resurgent growth dome": "*Application of hygromycin halts the growth of susceptible wild-type cells while resistant clones still present at the colony edge continue to expand outward in a semi-circular pattern (Fig. 1e). We will refer to these rapid expansions of resistant lineages after drug application as resurgent growth domes.*"
- We replaced all occurrences of "resurgent growth cone" (and variations thereof) with "resurgent growth dome"
- We fixed the typo in the original line 165, now line 194.

Comment 2L

Although the article is very well written, there are a few typos and minor grammatical mistakes. For example, "a compensatory mutation rates" (line 166); "level of at" (203); "extend of" (242); "stop to proliferate" (370).

We are delighted that the reviewer considers our article to be well written and thank her/him for pointing out remaining typos and minor grammatical mistakes, which we now fixed.

List of Changes:

- Line 196 in the new manuscript now reads “... a compensatory mutation at rate μ ...”
- Line 246 in the new manuscript now reads “...to level off at...”
- Line 285 in the new manuscript now reads “...the extent of...”
- Line 420 in the new manuscript now reads “...stop proliferating...”

Reviewer #3 (Remarks to the Author):

This manuscript presents a synthetic experimental model system for studying evolutionary rescue: the phenomenon where a previously less fit mutant gains an additional mutation that restores its fitness. A “resistant” strain of the yeast *S. cerevisiae* is engineered to be resistant to the antibiotic hygromycin B, but sensitive to the antibiotic cycloheximide (to which the parent strain is resistant), allowing the fitness of the resistant strain to be manipulated via addition of cycloheximide. The resistant strain is further engineered to randomly undergo a genetic switch to a state that is resistant to cycloheximide. The rate of this switch can be controlled by adding beta-estradiol.

Using this system, the authors investigate the dynamics of evolutionary rescue in spatially structured expanding colonies. Their key result relates to the balance between expansion of a clone due to the radial expansion of the growing front, and clone width-dependent selection, which they term “inflation-selection balance”.

I find this an impressive piece of work which reveals interesting insights into the effects of population spatial structure on evolution. My comments relate mainly to the interpretation of the results in terms of clinical relevance, and the clarity of some parts of the manuscript.

We thank the reviewer for the positive assessment of our work. The excellent comments helped us to further improve our manuscript in a number of critical points, which we address in detail below.

Comment 3A

1. The comparison to the “null model” in lines 177-182 is unclear. It is not explained at this stage what the null model is, or why it is called “null”. I guess it is the version of the diffusing-boundaries model without width dependent selection. But the reference is to figure 5, which actually shows a different version of the model, which does include width-dependent selection and agrees well with the experiments. This part should be made much clearer by actually explaining what the model is when it is introduced.

This is an excellent remark. As the reviewer correctly assumed, the null model is indeed the radial-random-walk model with constant selection. We refer to it as the “null” model as it represents the baseline to which our experimental results are compared.

We fully agree that a more detailed introduction of this model when it is first introduced would greatly facilitate the interpretation of our results. We reworked the passage in question to clarify the nature of our null model and the underlying rationale. We also included citations to previous studies that have investigated and validated this approach.

We kept the reference to Fig. 5a, as this panel depicts a generic illustration of the model, independent of the type of selection. Since the other panels in Fig. 5 indeed depict data from different selection modes of this model, including constant and width-dependent selection, we adjusted the caption to clarify what we consider to be the “null model” and which selection mode is displayed in which figure/panel.

We considered moving the illustration of the random-walk model to Fig. 2 but decided that it is much more valuable in the context of Fig. 5, where it is explored in full. However, we now indicate the link more clearly in both captions.

List of Changes:

- We now included an improved description of the null model when it is first introduced (lines 206 ff).
- We added references to previous implementations of this approach [ref. 19, 23 and 24].
- We updated the caption of Fig. 5a to now read: “*Schematic of radial-random-walk model, simulating clone boundaries as a selection-biased 1D random walk along a radially expanding periphery. Inset: Selection can either reflect the full fitness cost (null model, see Fig. 2b, f), be reduced by a constant factor (f) or change as a function of clone width ($b-g$).*”

Comment 3B

2. Frequently units are missing for r in the text (eg line 188 but also in many other places).

Thank you for pointing out this omission.

List of Changes:

- We included units for all values of r in the main text.

Comment 3C

3. In general the first results section would be clarified by separating the results for the compensated and uncompensated clones. For example in lines 195-203 it is not clear what type of clone is being discussed.

This is an excellent suggestion. We restructured our text accordingly.

List of Changes:

- We restructured the indicated results section to first describe the behavior of compensated clones (changes of mean width and width distributions) followed by a separate description of uncompensated clones (lines 226 ff).

Comment 3D

4. It should be clarified that “treatment failure” here refers to a specific context. Here for example there is no killing by the hygromycin B. The results would be different for an antibiotic that killed the bulk of the colony, freeing up space.

We agree. While yeast cells are indeed killed by hygromycin B, they remain structurally intact and do not free up any space. This might be potentially quite different from the scenario of a bacterial biofilm under antibiotic attack or a solid tumor undergoing chemotherapeutic treatment. We added a clarification of the specific context of our assay to the main text.

List of Changes:

- In line 145 ff we now added the following clarifying passage: *“It should be noted that the specific form of treatment failure is likely to be different in other contexts, such as the antibiotic treatment of a biofilm or the effects of chemotherapy on a solid tumor. For example, dead yeast cells retain their structural integrity upon treatment and do not free up any space.”*

Comment 3E

5. A claim is made that the results are very general, eg in lines 355-357 they are connected to bacterial biofilms and solid tumours. However the key findings depend crucially on two factors: radial expansion and width-dependent selection, that may not be relevant in bacterial biofilms. Bacterial biofilms do not necessarily expand laterally as they grow. Moreover the surfing mechanism that leads to width-dependent selection might not happen in biofilms where cells are quite sparse and surrounded by exopolysaccharide. I see this work as a very nice model study connecting biophysics, spatial structure and evolution, but I think the claims on general relevance should be toned down.

The reviewer is making an important point. The secretion of an EPS matrix, a common feature of bacterial biofilms, might indeed alter the behavior discussed in our work, adding an additional layer of complexity. For example, while we would expect that expansion driven by EPS secretion should exhibit similar long-range correlations in cell motion than expansion driven purely by cellular proliferation, we have not thoroughly explored this scenario. In addition, a different spatial growth mode, such as the absence of lateral expansion or a population growing in spatial confinement like a pore or a tube, might not exhibit the type of inflation considered in this work. We now discuss these limitations in greater detail in the discussion section of our work.

We agree that the extended claim of generality in our initial manuscript might have been misleading. Our intention is to highlight the minimal ingredients, i.e. inflation of the periphery and width-dependent selection due to collective motion, that is *sufficient* to produce the described effects and that these conditions are found in a wide range (though not all!) of natural compact populations. While we are still convinced that the fundamental physical effects of density driven expansion has the potential to result in inflation-selection balance in many other cellular populations, we agree

with the reviewer that we did not clearly point out the associated limitations of this notion and of our minimal model. We now acknowledge these limitations more clearly and toned down the claims on general relevance, as suggested.

List of Changes:

- We toned down the claim of generality throughout the manuscript by changing our wording in line (see e.g. line 394 ff.), now stating that width-dependent selection *can* inherently emerge in dense populations (rather than being an inevitable consequence).
- Lines 392 ff now reads: “*We previously demonstrated that such a clone-width dependent reduction in effective selection can inherently emerge as a result of collective cell dynamics in dense cellular populations*”
- Lines 401 ff now reads: “*However, the fundamental concept of inflation-selection balance and its evolutionary consequences may extend to other dense populations, including pathogenic bacterial biofilms or solid tumors, that exhibit the minimal set of necessary ingredients: i) the inflation of a peripheral growth layer and ii) width-dependent selection.*”
- We added a paragraph to our discussion, elaborating on the potential impact of EPS secretion, spatial confinement and ensuing limitations for the interpretation of our results (line 596 ff).
- In the discussion (line 635 ff), we now acknowledge the limits of our minimal model more clearly by stating: “*Yet, natural cellular populations exhibit many layers of complexity that are not captured by the minimal assumptions of our model and that might overlay, reshape or even eliminate inflation-selection balance and its effects on evolution.*”

Comment 3F

6. I found the text in lines 410-419, 439-448 and also 451-464 convoluted and hard to follow. These concepts could be explained more clearly.

We thank the reviewer for pointing out these hard-to-parse sections. We reworked the explanation of these concepts to improve clarity.

List of Changes:

- We improved our text to explain these complex concepts more clearly. The indicated sections correspond to the following new sections in the updated manuscript: lines 461 ff., 494 ff., and 511 ff.

In addition to the changes associated with reviewer comments, we fixed some minor inconsistencies:

List of additional Changes:

- We now consistently use “mean width” instead of a mixture of “mean/average width”
- We indicated the used double y-axis in Fig. 6k via color.
- We fixed additional typos

Reviewers' Comments:

Reviewer #1:

Remarks to the Author:

The authors have very carefully and adequately addressed all of my comments. The addition of a more thorough discussion of the parameters of the experimental system is an important addition that was implemented by the authors. I also think the updated manuscript title is better. I recommend publication.

Reviewer #2:

Remarks to the Author:

The authors have done a great job of responding to all my comments and queries. I've no further suggestions for improvement.

Reviewer #3:

None